# Short-term exposure to intermittent hypoxia leads to changes in gene expression seen in chronic pulmonary disease

Gang Wu[1†], Yin Yeng Lee[1,2†], Evelyn M Gulla[3], Andrew Potter[4], Joseph Kitzmiller[5], Marc D Ruben[1], Nathan Salomonis[2,6], Jeffery A Whitsett[5], Lauren J Francey[1], John B Hogenesch[1], David F Smith[3,7,8,9]*

[1]Divisions of Human Genetics and Immunobiology, Center for Circadian Medicine, Department of Pediatrics, Cincinnati Children's Hospital Medical Center, Cincinnati, United States; [2]Department of Pharmacology and Systems Physiology, University of Cincinnati College of Medicine, Cincinnati, United States; [3]Division of Pediatric Otolaryngology - Head and Neck Surgery, Cincinnati Children's Hospital Medical Center, Cincinnati, United States; [4]Division of Developmental Biology, Cincinnati Children's Hospital Medical Center, Cincinnati, United States; [5]Division of Pulmonary Biology, Cincinnati Children's Hospital Medical Center, Cincinnati, United States; [6]Division of Biomedical Informatics, Cincinnati Children's Hospital Medical Center, Cincinnati, United States; [7]Division of Pulmonary Medicine and the Sleep Center, Cincinnati Children's Hospital Medical Center, Cincinnati, United States; [8]The Center for Circadian Medicine, Cincinnati Children's Hospital Medical Center, Cincinnati, United States; [9]Department of Otolaryngology-Head and Neck Surgery, University of Cincinnati College of Medicine, Cincinnati, United States

*For correspondence:
david.smith3@cchmc.org

†These authors contributed equally to this work

Competing interests: The authors declare that no competing interests exist.

**Abstract** Obstructive sleep apnea (OSA) results from episodes of airway collapse and intermittent hypoxia (IH) and is associated with a host of health complications. Although the lung is the first organ to sense changes in oxygen levels, little is known about the consequences of IH to the lung hypoxia-inducible factor-responsive pathways. We hypothesized that exposure to IH would lead to cell-specific up- and downregulation of diverse expression pathways. We identified changes in circadian and immune pathways in lungs from mice exposed to IH. Among all cell types, endothelial cells showed the most prominent transcriptional changes. Upregulated genes in myofibroblast cells were enriched for genes associated with pulmonary hypertension and included targets of several drugs currently used to treat chronic pulmonary diseases. A better understanding of the pathophysiologic mechanisms underlying diseases associated with OSA could improve our therapeutic approaches, directing therapies to the most relevant cells and molecular pathways.

## Introduction

Obstructive sleep apnea (OSA) is a condition characterized by episodes of sleep-associated upper airway obstruction and intermittent hypoxia (IH). OSA occurs in approximately 2–5% of children (*Marcus et al., 2012*) and 33% of adults 30–69 years of age (*Benjafield et al., 2019*) in the USA. If untreated, OSA is associated with significant health consequences to the cardiovascular, neurological, and metabolic systems. Even young children with moderate to severe OSA can develop blood

pressure dysregulation (*Amin et al., 2004*), systemic hypertension (*Enright et al., 2003*; *Kohyama et al., 2003*), and left ventricular hypertrophy (*Amin et al., 2002*; *Amin et al., 2005*). OSA is associated with a significant socioeconomic burden in the USA (*Sullivan, 2016*). Despite available medical and surgical therapies, millions of children and adults with OSA are currently untreated or do not respond to available therapies. Cellular responses to changes in oxygen levels are primarily mediated by the hypoxia-inducible factors (HIFs). Although the lung is the first organ to sense large changes in inspired oxygen levels, little is known about the consequences of IH to the lung HIF-responsive pathways. Without a basic understanding of the molecular mechanisms that lead to diseases associated with IH and OSA, our ability to identify new treatments is significantly hindered.

Efforts to understand the effects of OSA have primarily focused on systemic inflammation (*Gozal et al., 2008*), oxidative stress (*Tauman et al., 2014*), and endothelial dysfunction (*Kheirandish-Gozal et al., 2013*; *Bhattacharjee et al., 2012*). However, the early causal events from IH exposure are not fully elucidated. Research is now focused on other possible pathogenic pathways that could be activated or suppressed in the presence of IH, leading to disease. For example, HIFs stabilized under low-oxygen conditions can affect the circadian transcriptional–translational feedback loop at the cellular level (*Adamovich et al., 2017*; *Kobayashi et al., 2017*; *Peek et al., 2017*; *Wu et al., 2017*; *Hogenesch et al., 1998*). Even acute exposure to IH results in dysregulation of the circadian clock that is time-of-day dependent and tissue specific, and these effects persist in some tissue for up to 24 hr after exposure (*Manella et al., 2020*). Pathways involved in immune responses and regulation can also be activated or suppressed in the presence of IH (*Cubillos-Zapata et al., 2017*; *Lam and Ip, 2019*), contributing to comorbid disease initiation and progression. Associations between IH and gene targets could be either pathogenic or protective responses for the lung. Additionally, the lung could be an effector rather than the target organ of IH, resulting in responses to IH that lead to multi-systemic effects.

While animal models of OSA have focused on physiologic responses to IH at organ and system levels, the determination of the contributions of individual cell types in the initiation and progression of disease has been challenging. Within organs, individual cells serve specific physiologic roles. As a result, pathways disrupted by stabilization of HIFs can affect cell types differently. Single-cell RNA sequencing (scRNA-seq) has emerged as a method for evaluating transcriptional states from thousands of individual cells (*Zhang et al., 2018*), advancing our understanding of how specific cell types contribute to physiology and disease (*Zhang et al., 2018*; *Plasschaert et al., 2018*).

In the present study, we used IH as a mouse model of OSA to better understand early cellular-specific consequences to the lung, the primary organ that first senses hypoxic episodes. We hypothesized that exposure to IH would lead to up- and downregulation of diverse expression pathways, that distinct cell populations would show distinctive responses to IH, and that changes in these gene expression pathways could provide therapeutic targets at the cell-specific level. We identify changes in both circadian and immune response pathways in lungs from mice exposed to IH. We also demonstrate strong similarities in the gene expression profiles from mice compared to those characteristics of human lung tissue from patients with diverse pulmonary diseases, including pulmonary hypertension and pulmonary fibrosis. Our results reveal potential candidates for cell-targeted therapy seeking to minimize effector responses of the lung that could lead to systemic disease. A better understanding of the pathophysiologic mechanisms underlying diseases associated with OSA could improve our therapeutic approaches.

## Results

### Short-term exposure to IH reshapes circadian and immune pathways in the lung

In humans, moderate to severe OSA is associated with interstitial lung disease with remodeling of the extracellular matrix (*Kim et al., 2017*). Lung is the primary organ that senses episodes of hypoxia and is therefore exposed to large fluctuations in the oxygen concentrations compared to other tissues throughout the body. For these reasons, we sought to identify initial changes in gene expression pathways in the lung in response to IH.

Mice were initially entrained to the same light:dark schedule to synchronize active and inactive phases. After 14 days of entrainment in the 12 hr:12 hr light:dark cycle, mice were exposed to IH or

room air (normoxia) for the entire 12 hr inactive phase for 9 days (*Figure 1A*). Hematoxylin and eosin (H and E) staining of whole lungs did not show comprehensive changes in architecture or inflammatory remodeling after exposure to IH (*Figure 1—figure supplement 1*). Bulk RNA sequencing (Bulk RNA-seq) was performed to explore transcriptomic effects of IH on lung tissue at the organ level. There were 374 genes (*Figure 1—figure supplement 2A*) upregulated and 149 downregulated in mouse lung after exposure to IH (Benjamini-Hochberg q value (BHQ) < 0.05 and fold change > 1.5). Not surprisingly, the top upregulated genes included well-known HIF-1 target genes (e.g. *Edn1*, *Bnip3*, and *Ankrd37*; *Figure 1—figure supplement 2B*).

We performed DAVID (*Huang et al., 2009*) enrichment analysis to identify biological processes associated with the top 200 up- and downregulated genes from mice exposed to IH. Pathways induced in response to hypoxia included circadian rhythm, angiogenesis, and extracellular matrix

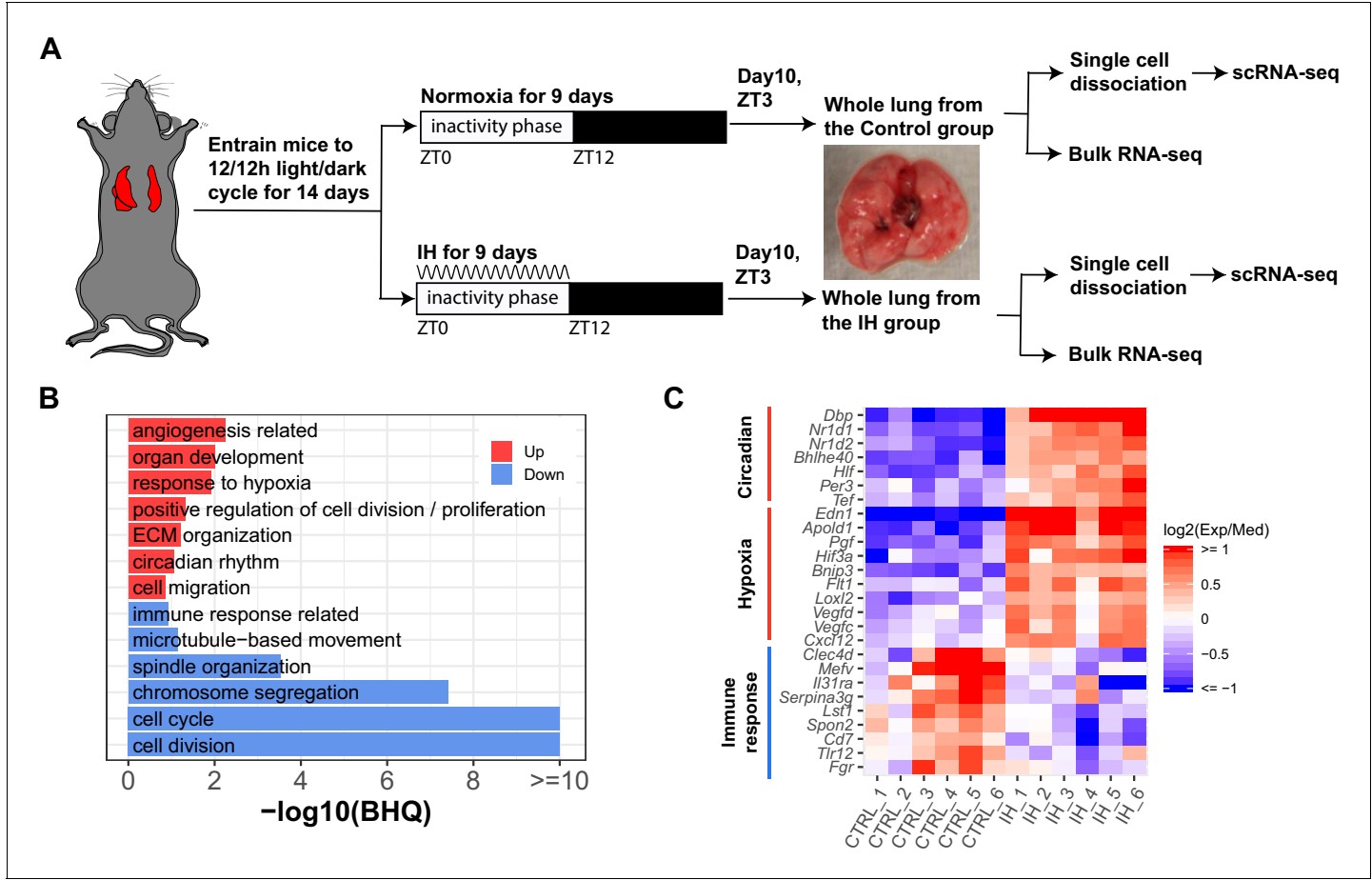

**Figure 1.** Short-term exposure to intermittent hypoxia reshapes circadian and immune pathways in the lung. (**A**) Schematic of IH protocol. Mice are entrained to the same 12 hr:12 hr light:dark cycle for 14 days prior to IH exposure. Mice are exposed to normoxia (controls) or intermittent episodes of hypoxia (21–6% oxygen saturation) followed by recovery to 21% oxygen over the entire 12 hr inactivity phase for 9 days. Mice are then sacrificed at ZT 3 (3 h after lights-on) on the 10th day for tissue harvest. Bulk RNA-seq and scRNA-seq were performed for each group. (**B**) Biological processes enriched in lung from mice exposed to IH vs controls. Enrichment analysis was performed in the DAVID database, using the top 200 up- and downregulated genes identified from differential expression analyses. Redundant biological processes are merged into one category. Biological processes enriched in up- and downregulated genes are indicated in red and blue bars, respectively. (**C**) The heatmap shows the fold change of associated genes in circadian rhythm, response to hypoxia, and immune response. The red and blue indicate up- and downregulated genes in the experimental group. There are six biological replicates for each group.

The online version of this article includes the following source data and figure supplement(s) for figure 1:

**Source data 1.** Numerical data for *Figure 1B,C*, *Figure 1—figure supplement 2*.

**Figure supplement 1.** Hematoxylin and eosin (HE)-stained sections of whole lung from mice exposed to IH vs controls do not reveal comprehensive histologic changes.

**Figure supplement 2.** Differentially expressed genes from bulk RNA-seq analysis of whole lung from mice exposed to IH vs controls.

organization (*Figure 1B,C*). As previously reported, there is a tight interaction between clock genes and HIFs (*Edgar et al., 2012*; *Gu et al., 2000*; *Hogenesch et al., 1998*; *McIntosh et al., 2010*; *Taylor and Zhulin, 1999*). Several circadian clock repressors (e.g. *Nr1d1*, *Nr1d2*, *Bhlhe40*, and *Per3*) were significantly upregulated in the IH group (*Figure 1C*). Increased expression of RNAs associated with angiogenesis, such as vascular endothelial growth factor (VEGF), was observed after IH, consistent with findings in a mouse model of prolonged exposure to IH (*Reinke et al., 2011a*) and after 72 hr of IH exposure to endothelial cells in vitro (*Wohlrab et al., 2018*). There is a tight relationship between angiogenesis and pulmonary hypertension (*Tuder and Voelkel, 2002*), a clinical consequence associated with OSA. Unexpectedly, RNAs associated with immune responses and cell cycle were significantly downregulated after 9 days of IH (*Figure 1B,C*). Present findings contrast with the general concept that HIFs are important regulators of inflammation and immune responses (*Eltzschig and Carmeliet, 2011*; *Scholz and Taylor, 2013*; *Taylor et al., 2016*). For example, activation of neutrophils by HIFs is largely considered proinflammatory (*Walmsley et al., 2005*; *Peyssonnaux et al., 2005*; *Taylor et al., 2016*). The cell cycle and cell division downregulation may be associated with HIFs induced in cell cycle arrest in response to IH (*Koshiji et al., 2004*).

## Single-cell sequencing identifies 19 distinct cell types in the lungs of IH and control mice

We detected significant differences related to a number of biological functions at the tissue level. We then applied single-cell transcriptomics to identify cell type-specific effects of IH. We performed three biological replicates in each group to improve the statistical power in differential gene expression analysis at the single cell level. In total, we sequenced 12,324 and 16,125 pulmonary cells from IH and control mice, with 3542–5641 cells per biological replicate. Unsupervised analysis identified 25 transcriptionally distinct cell clusters, corresponding to 19 distinct cell types (*Figure 2A,B*) based on the expression of established marker genes (see Materials and methods), including stromal, epithelial, endothelial, immune, and small numbers of other cell types. In brief, AltAnalyze identified 40–60 marker genes for each cell cluster. We annotated each cluster to a cell type using enrichment analysis between these marker genes and a comprehensive reference marker gene list, which is collected from public databases and published scRNA-seq studies performed in mouse or human lungs. As shown in *Figure 2C*, the cell-type assignment was also validated with multiple well-known cell-type markers. The proportion of endothelial, alveolar epithelial type II (AT2), and fibroblast/myofibroblast cells were modestly increased, but the proportion of immune cells (e.g. B and T cells) was decreased in the IH-exposed mice (*Figure 2—figure supplement 1*). Overall, the proportional variation of lung cell types was small (BHQ > 0.05). We further performed confocal immunofluorescence to evaluate variation in lung structure and to quantify the differences in cell number of major cell types in mice exposed to IH compared to controls.

## Short-term exposure to IH did not lead to comprehensive histologic changes in the lung

Similar to H and E histology, confocal immunofluorescence microscopy did not demonstrate comprehensive inflammatory remodeling. However, some modest changes were noted with specific cell populations. Immunostaining for endothelial markers FOXF1 and LYVE1 showed modest increases in the number of endothelial cells in mice exposed to IH (*Figure 3C,D*) compared to control mice (*Figure 3A,B*). Also important, staining for MKI67 did not show statistically significant changes in cell proliferation (*Figure 3E–H*, *Figure 3—figure supplement 1*). Expression levels of HOPX and SFTPC in alveolar type I and II cells, respectively, were not different for IH (*Figure 3K,L*) versus control (*Figure 3I,J*) mice. The trend for small increases in AT2 cells, although not significant, was also seen from the scRNA-seq data (*Figure 2—figure supplement 1*, *Figure 3—figure supplement 1*). Immunostaining for the progenitor marker SOX9 (*Figure 3I–L*) or the extracellular matrix marker POSTN (*Figure 3M–P*) did not demonstrate any significant changes after IH exposure. Overall, there was a modest increase (11.5%; BHQ < 0.05) of endothelial cells in mice exposed to IH compared to controls (*Figure 3—figure supplement 1*); however, this does not necessarily reflect a biologic difference. This trend for an increasing number of endothelial cells in the histologic samples was similar to the changes in cell numbers seen from the scRNA-seq data (*Figure 2—figure supplement 1*, *Figure 3—figure supplement 1*). Other cell percentages, including AT2 cells and proliferating cells,

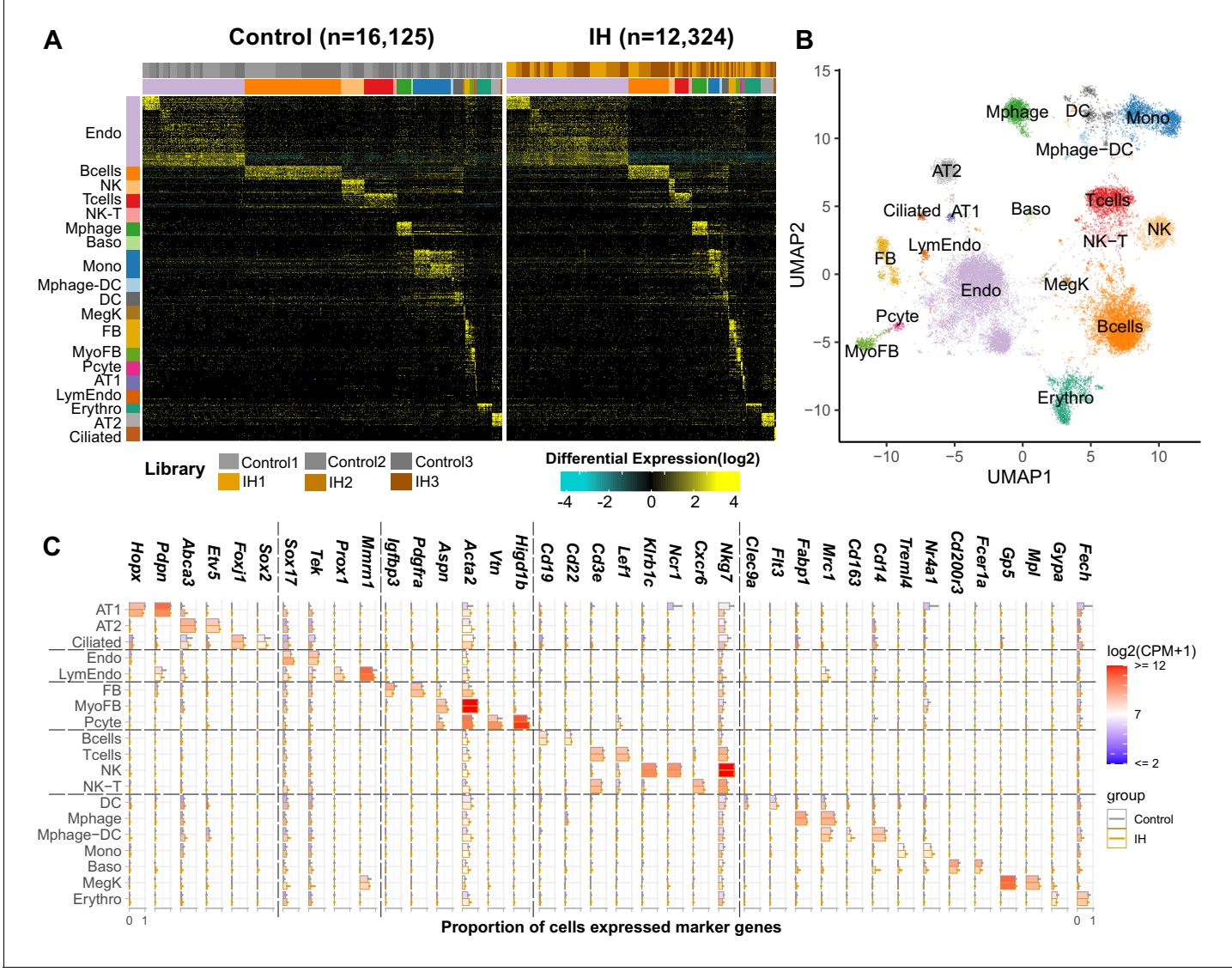

**Figure 2.** Single-cell sequencing identifies 19 distinct cell types in the lungs of intermittent hypoxia and control mice. (A) Heat map of AltAnalyze selected marker genes for each cell population. Columns and rows represent individual cells and marker genes, respectively. Biological replicates and cell types are indicated by the color bars on the top. cellHarmony was used to align cells from control to IH groups. (B) UMAP projection of whole lung cell populations from IH and control mice in the heatmap. (C) Expression of marker genes for each cell type from experimental and control mice. Error bars indicate standard deviation of cell proportion from the three replicates. The list of cell types include endothelial cells (Endo), B cells (Bcells), natural killer cells (NK), T cells (Tcells), natural killer T cells (NK-T), macrophages (Mphage), basophils (Baso), monocytes (Mono), macrophages-dendritic CD163[+] cells (Mphage-DC), dendritic cells (DC), megakaryocytes (MegK), fibroblasts (FB), myofibroblasts (MyoFB), pericytes (Pcyte), alveolar epithelial type I cells (AT1), lymphatic endothelial cells (LymEndo), erythroblasts (Erythro), alveolar epithelial type II cells (AT2), and ciliated cells (Ciliated). CPM indicates UMI count per million.

The online version of this article includes the following source data and figure supplement(s) for figure 2:

**Source data 1.** Numerical data for *Figure 2*, *Figure 2—figure supplement 1*.
**Figure supplement 1.** Percentage of detected lung cell types from mice exposed to IH vs controls.

were not significantly different between IH and control groups. We did not see comprehensive changes in alveolar area and alveolar wall thickness based on morphometric quantification for mice exposed to IH compared to controls (*Figure 3—figure supplement 2A–C*). The average MaxFeret90 measurements and alveolar areas were 21.0 µM and 419 µM² in mice exposed to IH, while the average MaxFeret90 measurements and alveolar areas were 23.5 µM and 564 µM² in control mice. The mean alveolar wall thickness was 4.81 µM for mice exposed to IH and controls.

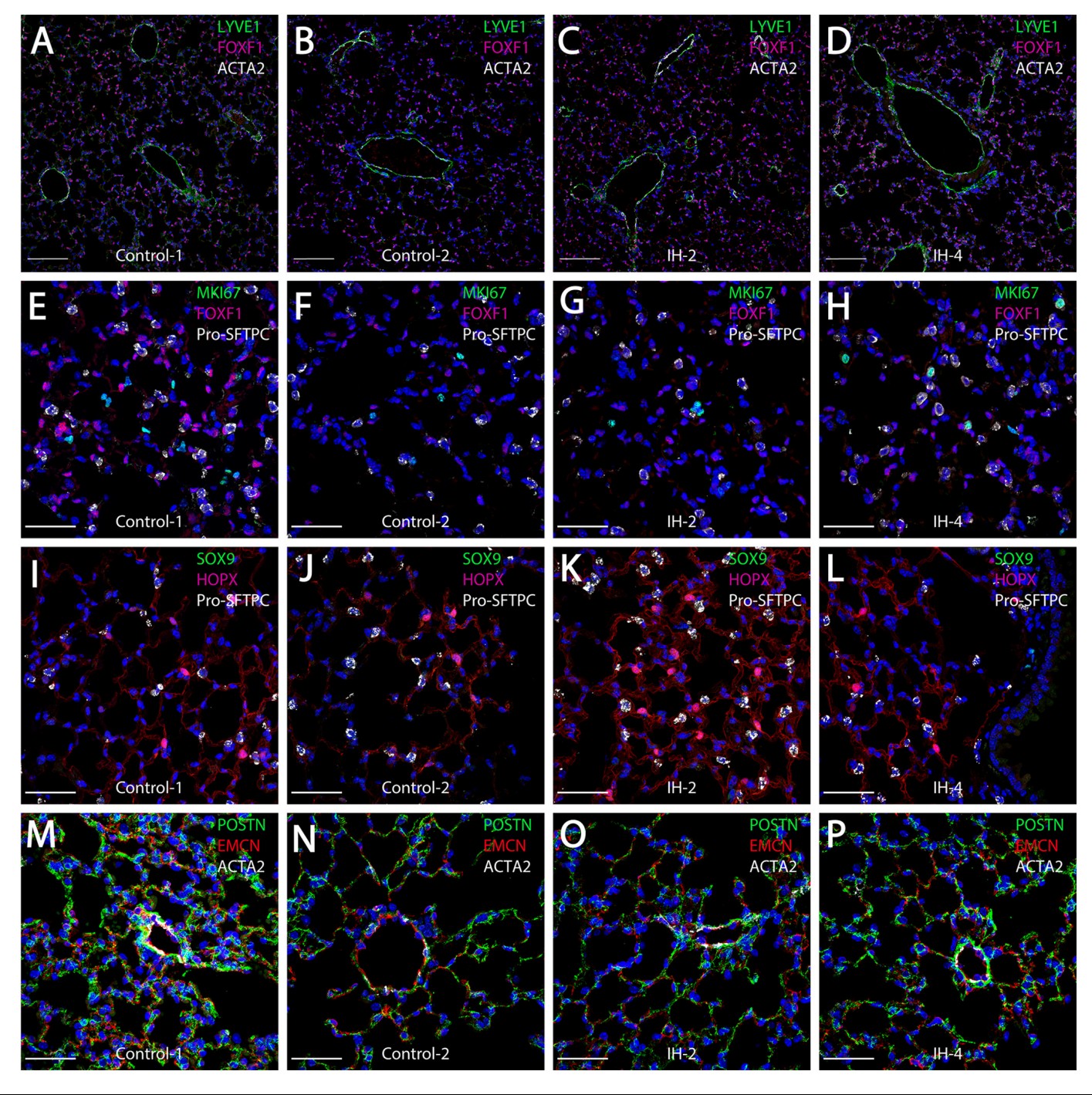

**Figure 3.** Short-term exposure to intermittent hypoxia did not lead to comprehensive histologic changes in the lung. (A–D) Immunostaining for LYVE1, FOXF1, and ACTA2: LYVE1 is expressed at high levels in lymphatic and vascular endothelial cells. FOXF1 and ACTA2 are expressed at high levels in vascular endothelial and smooth muscle cells, respectively. (E–H) Immunostaining for MKI67, FOXF1, and Pro-SFTPC: MKI67 is a marker for cell proliferation. Pro-SFTPC shows high expression levels in alveolar epithelial type II cells. (I–L) Expression levels of SOX9, HOPX, and Pro-SFTPC: SOX9 is a marker for progenitor cells. HOPX shows high expression levels in the alveolar epithelial type I cells. (M–P) Immunostaining for POSTN, EMCN, and ACTA2: POSTN is used to stain extracellular matrix. EMCN is a marker for capillaries and veins/venules. The scale bars for (A–D) represent 100 μm. The scale bars for (E–P) represent 40 μm.

The online version of this article includes the following source data and figure supplement(s) for figure 3:

**Source data 1.** Numerical data for *Figure 3—figure supplements 1* and *2*.

**Figure supplement 1.** The cell percentages for endothelial, AT2, and proliferating cells in mice lung exposed to IH vs controls.

*Figure 3 continued on next page*

*Figure 3 continued*

**Figure supplement 2.** Alveolar area and alveolar wall thickness are quantified from confocal images for mice exposed to IH compared to controls.

## Diverse expression pathways were up- and downregulated in the presence of IH

We further explored the early cell type-specific response to IH in mouse lung by aggregating single-cell data into 'pseudo-bulk' data to compare biological replicates for each identified cell type (see Materials and methods for details). For each biological replicate, we summed the reads for each gene in the same cell type. Total read counts between replicates and the two conditions were normalized by size factors in DESeq2 to reduce the impact of cell number differences. Using DESeq2 (*Love et al., 2014*), the number of up- or downregulated genes in different lung cell types in response to IH were not equal at the same statistical cutoff (*Figure 4—figure supplement 1*). For example, there were 607 and 550 genes up- and downregulated in endothelial cells, while there were 186 and 139 genes up- and downregulated in B cells at p<0.05. We selected the top 200 up- and downregulated genes (ranking by the p-value) from each cell type in the pathway enrichment analysis to balance the input gene number difference. From the DAVID enrichment analysis, diverse biological processes were up- and downregulated in different cell types in response to IH (*Figure 4A–C*, *Figure 4—figure supplement 2A,B*). For example, hypoxia-responsive and circadian pathways were enriched in those upregulated genes in response to IH in endothelial cells, myofibroblasts, and AT2 cells. Surprisingly, circadian pathways were highly enriched in multiple cell populations, not just epithelial cells, a population that is important for circadian rhythmicity in the lung (*Gibbs et al., 2009*). The co-upregulation of genes in these two pathways suggest that the genome-wide co-regulation of hypoxia and the circadian pathway is tighter in lung endothelial, myofibroblast, and AT2 cells than other cell types.

Our data demonstrate that cell adhesion genes are upregulated in endothelial cells, myofibroblasts, fibroblasts, and pericytes after exposure to IH. These results suggest that lung endothelial and stromal cells may contribute to the increased circulating cell adhesion molecules in OSA patients. Interestingly, cell adhesion molecules that control leukocyte trafficking are increased in OSA patients (*Ohga et al., 1999*) and reduced after continuous positive airway pressure (CPAP) treatment (*Chin et al., 2000*; *Pak et al., 2015*). Immune response-related and antigen processing and presentation were enriched among those downregulated genes in monocytes, macrophages-dendritic cells, NK cells, and erythroblasts. The decrease among immune related genes in these cell types is consistent with the bulk RNA-seq data. It is known that a hypoxic tumor microenvironment will induce immunosuppression (*Sitkovsky et al., 2008*). Hypoxia may also cause immunosuppression in patients with OSA and contribute to the increased incidence of lung cancer in this population (*Campos-Rodriguez et al., 2013*).

As expected from specific genes in each biological process, the response level of the same genes were different in multiple cell types (*Figure 4B,C*). For example, the circadian repressor gene, *Nr1d1*, was more responsive to IH in endothelial and AT2 cells than lymphatic endothelial cells and AT1 cells. We also noted cell-specific responses for the downregulated genes. For example, immune response genes (e.g. *Iglc3*, *S100a8*, and *Oas3*) decreased more in monocytes than macrophages in response to IH. These results suggest that a cell type-specific response to short exposures of IH exists at a single-gene level, findings that may be related to the progression of OSA-associated chronic pulmonary diseases. For example, bleomycin-induced fibrosis is induced in mice by an *Nr1d1* mutation in specific lung cells (*Cunningham et al., 2020*).

## Pulmonary vascular endothelial subpopulations show distinctive responses to IH

Recent studies show distinct vascular endothelial cell subpopulations in mouse and human lung (*Gillich et al., 2020*). Our vascular endothelial populations were annotated to endothelial artery, vein, capillary aerocytes (Cap-a), and general capillary (Cap-g) cells (*Figure 5A*, *Figure 5—figure supplement 1*). Interestingly, we found that endothelial cells demonstrated profound changes in gene expression profiles in response to IH. The endothelial capillary cells were more responsive to IH compared to endothelial artery and vein cells (*Figure 5B*). For example, at BHQ < 0.2, more than

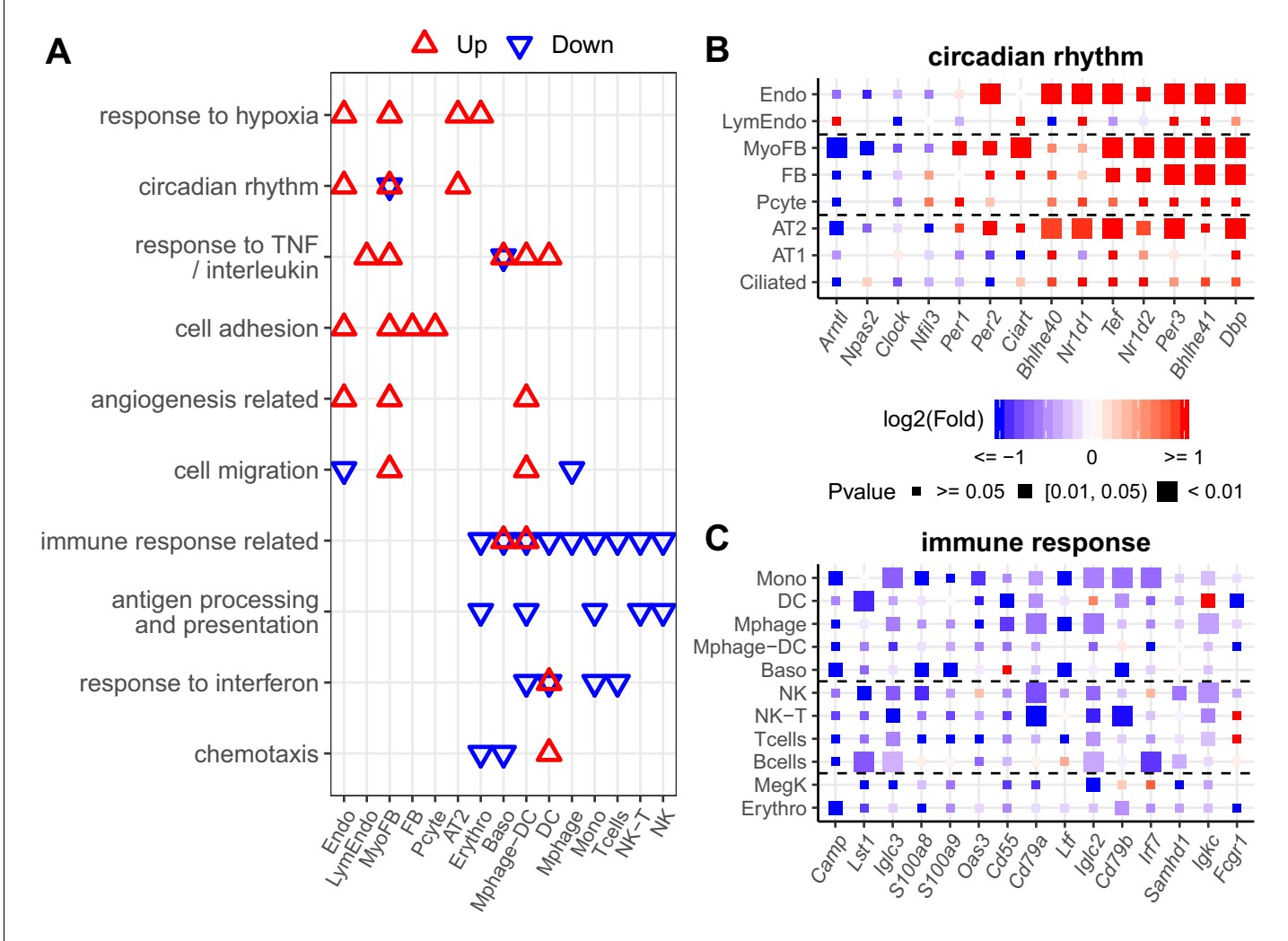

**Figure 4.** Diverse expression pathways were up- and downregulated in the presence of intermittent hypoxia. (**A**) Biological processes enriched in different cell types from lungs of mice exposed to IH vs controls. Enrichment analysis was performed in the DAVID database, using the top 200 up- and downregulated genes identified from each cell type. Redundant biological processes are merged into one category. Biological processes enriched in up- and downregulated genes are indicated in red and blue triangles, respectively. (**B**) Expression variation of well-established genes involved in circadian rhythm for endothelial, epithelial, and mesenchymal cells. (**C**) Expression variation of well-established genes involved in immune response for immune-associated cells. For (**B**) and (**C**), the fold change is indicated by the color, and the p-value for differential expression is indicated by the point size. The list of cell types include: endothelial cells (Endo), B cells (Bcells), natural killer cells (NK), T cells (Tcells), natural killer T cells (NK-T), macrophages (Mphage), basophils (Baso), monocytes (Mono), macrophages-dendritic CD163[+] cells (Mphage-DC), dendritic cells (DC), megakaryocytes (MegK), fibroblasts (FB), myofibroblasts (MyoFB), pericytes (Pcyte), alveolar epithelial type I cells (AT1), lymphatic endothelial cells (LymEndo), erythroblasts (Erythro), alveolar epithelial type II cells (AT2), and ciliated cells (Ciliated).

The online version of this article includes the following source data and figure supplement(s) for figure 4:

**Source data 1.** Numerical data for *Figure 4*, *Figure 4—figure supplements 1* and *2*.
**Figure supplement 1.** p-value distribution of the top 200 up- and downregulated IH-responsive genes in 19 lung cell types.
**Figure supplement 2.** Enriched biological processes from differentially expressed genes from whole mouse lungs exposed to IH vs controls.

100 genes were significantly upregulated in aerocytes and general capillary cells. However, only one gene in the arterial endothelial cells and 57 genes in the venular endothelial cells were significantly upregulated at the same cutoff. This trend persisted at other BHQ cutoffs (*Figure 5—figure supplement 2*). Given the location of lung capillary cells around the alveoli and positioned at the air–blood barrier, these results indicate that the capillary cells may be among the first lung endothelial cell groups that respond early after initial exposure to IH. Among the capillary cells, general capillary

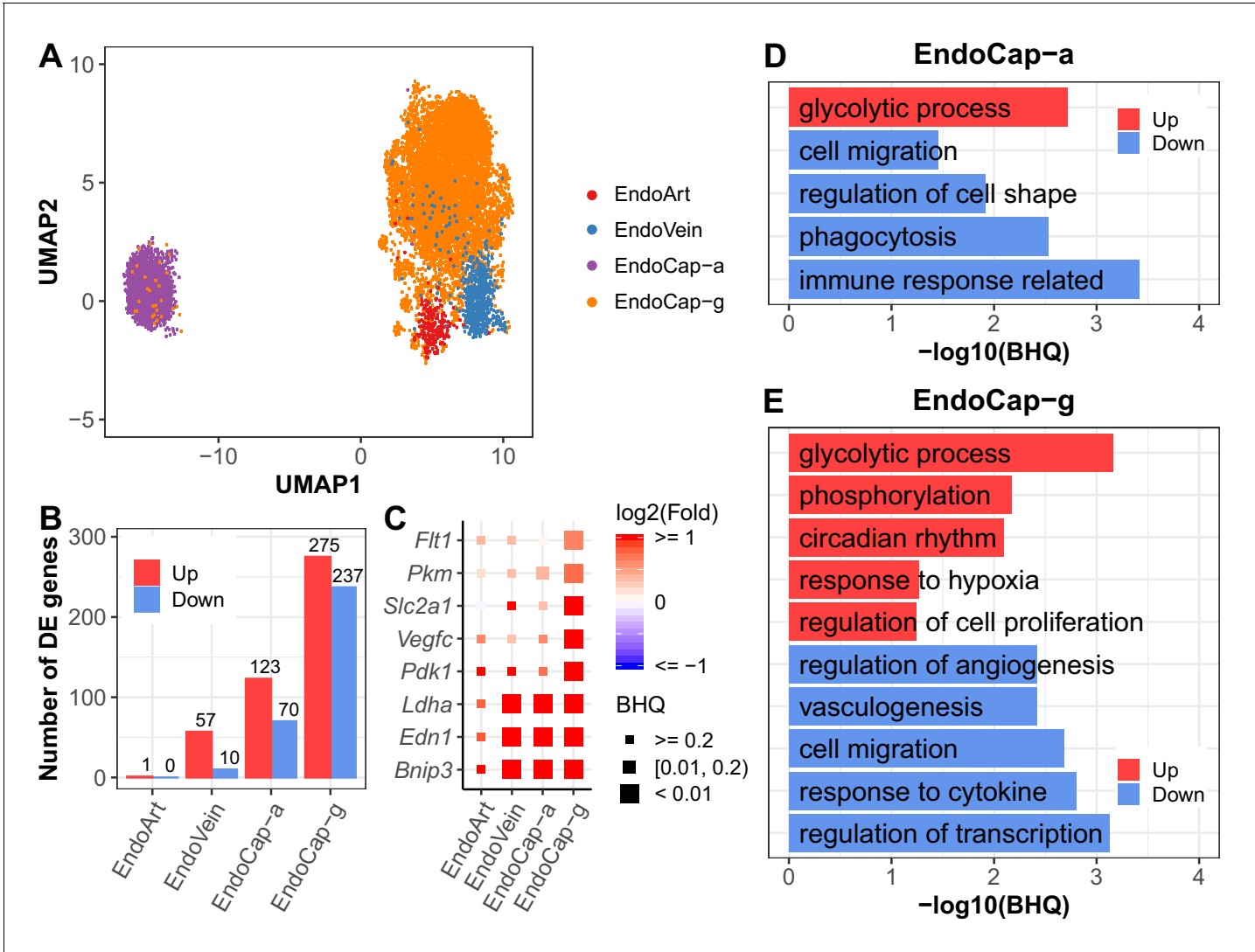

**Figure 5.** Pulmonary vascular endothelial subpopulations show distinctive responses to intermittent hypoxia. (**A**) UMAP projection of cells from four lung endothelial subpopulations. There are 22 cells outside the range of this figure. (**B**) The number of differentially expressed (BHQ < 0.2) genes in each endothelial subpopulation from IH vs controls. (**C**) Expression variation of related genes in response to IH is shown for each endothelial subpopulation. Biological processes enriched in aerocytes (**D**) and general capillary cells (**E**) from IH vs control mice. Enrichment analysis was performed in the DAVID database, using up- and downregulated genes (BHQ < 0.2). Redundant biological processes are merged into one category. Top five biological processes enriched in up and downregulated genes are indicated in red and blue bars, respectively. The list of endothelial subpopulations include endothelial artery (EndoArt), vein (EndoVein), general capillary (EndoCap-g) cells, and aerocytes (EndoCap-a).

The online version of this article includes the following source data and figure supplement(s) for figure 5:

**Source data 1.** Numerical data for *Figure 5*, *Figure 5—figure supplements 1–3*.
**Figure supplement 1.** Validation of endothelial subpopulation assignments with known markers.
**Figure supplement 2.** Number of differentially expressed IH-responsive genes in four lung endothelial subpopulations using BHQ cutoffs.
**Figure supplement 3.** Predicted key transcription factors (TFs) for the endothelial capillary subtypes.

cells are more responsive to hypoxia than aerocytes. More genes were significantly up and downregulated in general capillary cells than aerocytes (*Figure 5B*). Hypoxia-responsive genes (e.g. *Pdk1*, *Vegfc*, *Slc2a1*, *Pkm*, and *Flt1*) showed higher levels of expression variation (fold change and significance) in general capillary cells than aerocytes (*Figure 5C*), demonstrating variation at the subpopulation level. Aerocytes are specialized for gas exchange, and general capillary cells function in capillary homeostasis and aerocyte production during repair (*Gillich et al., 2020*). Our observed response from general capillary cells may be related to its role in capillary homeostasis. Alternatively,

proximity and interaction with other cell types, such as fibroblasts or immune cells, may also help to explain these findings. For example, general capillary cells contact stromal cells while aerocytes lie in close proximity to AT1 cells (*Gillich et al., 2020*).

We further compared the response of general capillary cells and aerocytes to IH at the pathway level. Glycolytic process was upregulated and cell migration was downregulated in both aerocytes and general capillary cells in response to IH (*Figure 5D,E*). Given the association between glycolysis, cytoskeletal remodeling, and cell migration in other cell types (*Shiraishi et al., 2015*), the similar enrichment trends for these pathways is not surprising. Additionally, without vascular growth associated with chronic IH, glycolysis may be used to meet metabolic demand. Interestingly, changes in the glycolytic process were specific to endothelial cell types (*Figure 4—figure supplement 2*). On the other hand, aerocytes and general capillary cells demonstrated more differences in enrichment pathways (*Figure 5D,E*). For example, regulation of cell proliferation and regulation of angiogenesis were only enriched for those up- and downregulated genes in general capillary cells in response to IH. These pathways are tightly correlated with its specific role in capillary homeostasis as a capillary progenitor cell and in regulation of vasomotor tone (*Gillich et al., 2020*). While phagocytosis, immune response, and regulation of cell shape were only enriched for those downregulated genes in aerocytes (*Figure 5D,E*), this may reflect aerocyte's specific role in leukocyte trafficking and gas exchange, both highly influenced by exposure to IH.

We used SINCERA (*Guo et al., 2015*) to predict key transcription factors (TFs) in the inferred transcriptional regulatory network in general capillary and aerocytes (*Figure 5—figure supplement 3A,B*; Materials and methods). Hypoxia-associated TFs (e.g. *Epas1*, *Jun*, and *Ahr)*, *Foxf1* (canonical endothelial cell marker), and *Ybx1* (glycolysis related) were among the top-ranked TFs in both aerocytes and general capillary cells. *Tbx2*, *Tbx3*, and *Meox1* were among the top-ranked TFs only in aerocytes, which are considered aerocyte-specific TFs (*Gillich et al., 2020*). Interestingly, three immune response-related TFs (*Hlx*, *Rarg*, and *Smarca2*) were only top-ranked in aerocytes. Additionally,*Tcf4* and *Sox17* were among the top-ranked TFs only in general capillary cells, which are associated with endothelial cell regeneration or stem cell self-renewal (*van Es et al., 2012*; *Liu et al., 2019*). *Elk3* and *Casz1* were among the top-ranked TFs in general capillary cells and are related to angiogenesis (*Heo and Cho, 2014*; *Charpentier et al., 2013*). The pathway-level differences between aerocytes and general capillary cells may be driven by these predicted top-ranked TFs, which need further experimental validation.

Evaluation of the expression pathways at the single-cell resolution demonstrated significant changes in multiple cell types. With this information, we then wanted to identify potential candidates for therapeutic intervention in response to IH.

## Pulmonary disease-regulated genes provide clinical implications for OSA at the cell-specific level

OSA is associated with an array of pulmonary diseases, such as interstitial lung disease (ILD) (*Kim et al., 2017*), idiopathic pulmonary fibrosis (IPF) (*Lancaster et al., 2009*), and pulmonary hypertension (PH) (*Sajkov and McEvoy, 2009*; *Chaouat et al., 1996*). IH led to significantly more upregulated than downregulated pulmonary disease-associated genes (*Figure 6A*, *Figure 6—figure supplement 1*). IH-induced expression in myofibroblasts and fibroblasts demonstrated enrichment of genes associated with pulmonary fibrosis (PF). However, the disease genes were not equally expressed or upregulated in these cell types (*Figure 6B*). For example, *Ptgis* is a PH-associated gene highly expressed in myofibroblasts and fibroblasts compared to other cell types (*Figure 6—figure supplement 2*). *Ptgis* is also a target gene for epoprostenol – a drug used for treating PH (*Sitbon and Vonk Noordegraaf, 2017*). The IPF-associated gene, *Thbs1* (*Idell et al., 1989*; *Kuhn and Mason, 1995*; *Yehualaeshet et al., 2000*), was highly expressed and more responsive to IH in myofibroblasts compared to other cell types. *THBS1* is upregulated in lung stromal cells of patients with chronic interstitial lung diseases (e.g. IPF) compared to healthy controls (Gene Explorer under Banovich/Kropski in http://www.ipfcellatlas.com/; *Habermann et al., 2020*). *Msr1* (*Silverman et al., 2002*; *Hersh et al., 2006*) is a chronic obstructive pulmonary disease (COPD)-associated gene, which was highly expressed in macrophage-DCs, basophils, and monocytes, consistent with scRNA-seq data from patients with IPF (Gene Explorer under Banovich/Kropski in http://www.ipfcellatlas.com/; *Habermann et al., 2020*). These data highlight the similarity of the IH-associated signatures with cell type-specific responses in an array of pulmonary diseases.

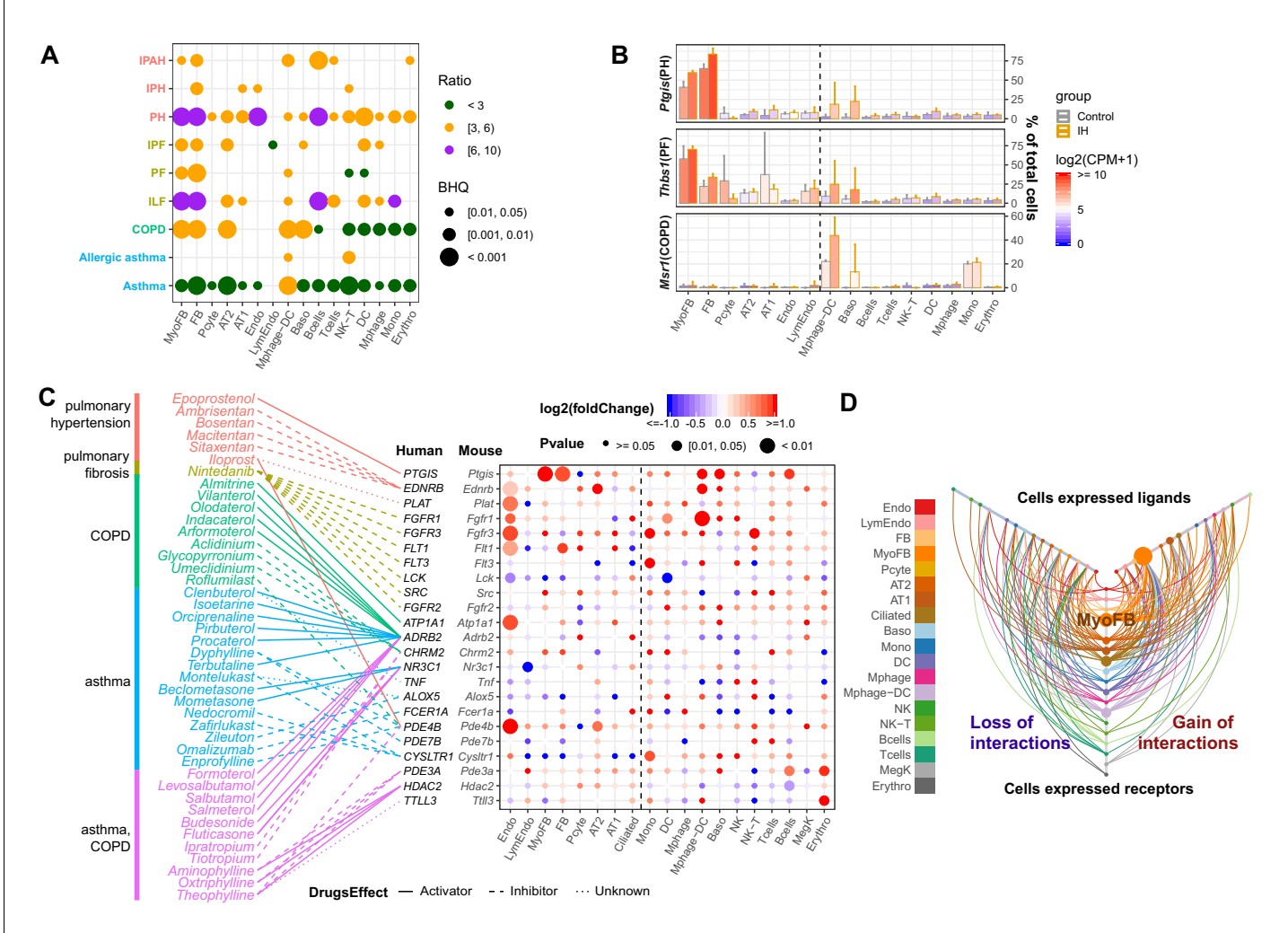

**Figure 6.** Pulmonary disease-regulated genes provide clinical implications for OSA at the cell-specific level. (A) Pulmonary disease-associated genes are enriched in upregulated genes in different cell types from whole lungs of mice exposed to IH vs controls. Enrichment analysis was performed in the DisGenet human database, using the top 200 upregulated genes identified from each cell type. The enrichment ratio is indicated by color. The point size indicates the enrichment BHQ from a Fisher's exact test. The pulmonary diseases include asthma, allergic asthma, chronic obstructive airway disease (COPD), interstitial lung fibrosis (ILF), pulmonary fibrosis (PF), idiopathic pulmonary fibrosis (IPF), pulmonary hypertension (PH), idiopathic pulmonary hypertension (IPH), and idiopathic pulmonary arterial hypertension (IPAH). (B) The disease-associated genes vary in expression level and percentage of cells that express those genes. The fold change is indicated by the color, and the percentage of cells that express those genes is indicated by the height of the bar. Control and experimental groups are indicated by gray and orange box borders, respectively. (C) Dozens of pulmonary drug targets show differential expression in multiple cell types in lungs from mice exposed to IH. Drug classes used to treat different pulmonary diseases are indicated by text color. Drug effect is indicated by the line type. Fold change is indicated by the point color, and the p-value of differential expression is indicated by the point size. (D) The hive plot shows ligand–receptor interaction changes between pairs of cell types in response to IH. Nodes indicate cell-expressed ligands (horizontal axis) or receptors (vertical axis). Size of nodes are in proportion to the number of interactions changed for the cell type. Width of lines show numbers of interactions gained (right) or lost (left) between the pairs of cell types. The list of cell types include endothelial cells (Endo), B cells (Bcells), natural killer cells (NK), T cells (Tcells), natural killer T cells (NK-T), macrophages (Mphage), basophils (Baso), monocytes (Mono), macrophages-dendritic CD163+ cells (Mphage-DC), dendritic cells (DC), megakaryocytes (MegK), fibroblasts (FB), myofibroblasts (MyoFB), pericytes (Pcyte), alveolar epithelial type I cells (AT1), lymphatic endothelial cells (LymEndo), erythroblasts (Erythro), alveolar epithelial type II cells (AT2), and ciliated cells (Ciliated).

The online version of this article includes the following source data and figure supplement(s) for figure 6:

**Source data 1.** Numerical data for *Figure 6*, *Figure 6—figure supplements 1–5*.

**Figure supplement 1.** Pulmonary disease-associated genes are enriched in downregulated genes in different cell types in the lung from mice exposed to IH.

**Figure supplement 2.** Pulmonary drug targets with cell type-specific expression profiles in mouse lung.

*Figure 6 continued on next page*

*Figure 6 continued*

**Figure supplement 3.** Ligand–receptor interaction analysis reveals a major role for myofibroblasts in activating the FGF signaling pathway as a response to IH.

**Figure supplement 4.** Enriched REACTOME gene sets for lung myofibroblast cells in mice exposed to IH vs controls.

**Figure supplement 5.** Prediction of key transcription factors (TFs) for the myofibroblast.

The top 200 up- or downregulated genes in each cell type exposed to IH were linked with drug target genes extracted from DrugBank (*Wishart et al., 2008*). We obtained 42 drugs used as activators or inhibitors of 23 target genes to treat pulmonary hypertension, pulmonary fibrosis, COPD, and asthma (*Figure 6C*). Interestingly, nearly one third of these drug target genes were more responsive to IH in endothelial cells compared to other cell types (*Figure 6C*, *Figure 6—figure supplement 1*). We further evaluated changes in the interactions of these drug-targeted disease genes in each cell type using CellPhoneDB (*Vento-Tormo et al., 2018*). There were 32 ligand–receptor interactions found in the database using these drug-targeted disease genes. Overall, myofibroblasts were involved in 98 of 176 gains of interaction (BHQ = 4.28e-50), demonstrating the significance of this cell type in early IH-associated responses (*Figure 6D*, *Figure 6—figure supplement 3*). Here, we showed the importance of myofibroblasts in activating the fibroblast growth factor (FGF) signaling pathway (*Figure 6—figure supplement 3*). We then performed further analysis on myofibroblasts. The GSEA showed upregulation of multiple extracellular matrices (ECM)-associated pathways (e.g. ECM proteoglycans, non-integrin membrane ECM interactions, and collagen chain trimerization) in myofibroblasts (*Figure 6—figure supplement 4*) after exposure to IH. FGF signaling pathway has a tight interaction with ECM (*Taipale and Keski-Oja, 1997*; *Ornitz and Itoh, 2015*), which may be associated with activation of the FGF signaling pathway in myofibroblasts in response to IH. In the inferred transcriptional regulatory network, the majority of the 25 top-ranked TFs were shared between the control and IH groups (*Figure 6—figure supplement 5*). Interestingly, only *Dbp* and *Hlf* were top-ranked TFs in mice exposed to IH (*Figure 6—figure supplement 5*). These results suggest that IH-induced circadian dysregulation may drive the transcriptional changes in myofibroblasts.

## Discussion

OSA results from intermittent episodes of airway collapse and hypoxemia and is associated with dementia (*Osorio et al., 2015*), diabetes (*Punjabi and Beamer, 2009*), hypertension (*Marin et al., 2012*), heart failure (*Gottlieb et al., 2010*), and stroke (*Valham et al., 2008*). How cellular responses to IH and hypoxemia initiate and cause disease progression in multiple organs remains unknown. Using IH as a mouse model of OSA, we show profound, cell type-specific changes in genome-wide RNA expression in the lung. RNA profiles from lungs of mice exposed to IH shared similarity with gene expression changes in human lung from patients with pulmonary disease, including PH, COPD, and asthma. OSA is associated with injury to alveolar epithelial cells and extracellular matrix remodeling, key features of ILD (*Kim et al., 2017*). Although it is known that pulmonary diseases share general mechanisms, such as systemic inflammation and oxidative stress (*McNicholas, 2009*), there is an incomplete understanding of the early-stage changes in the lung from OSA.

In the present study, macrophages, dendritic cells, and NK cells were the only populations to demonstrate altered oxidation–reduction in the early stages of IH exposure. It was previously demonstrated that chronic IH in mice caused the release of free oxygen radicals in the lung (*Tuleta et al., 2016*), effects that could be blunted with antioxidative agents (*Tuleta et al., 2016*). PH from chronic IH was associated with enhanced NADPH oxidase expression, and knockout mice lacking one of these subunits demonstrated attenuated effects of chronic IH (*Nisbet et al., 2009*). Which cells mediate these effects in the lung? This type of knowledge can help direct therapeutics to the most relevant cells and molecular pathways.

Our data provide insight into the early cellular responses that drive disease progression in OSA. By identifying the roles of individual cells in disease, we have the opportunity to test targeted therapeutics, focusing specifically on the most pathologically relevant cells and molecular pathways. Using these data, we further explored the possibility of selecting novel drug target candidates in other cell types. Studies using scRNA-seq are already being used to identify novel cell populations in disease. For example, *Xu et al., 2016* identified a loss of normal epithelial cells in the development of IPF. In

another study using single-cell profiling of bronchial epithelial cells, the major source of cystic fibrosis transmembrane conductance regulator (CFTR) activity, the pulmonary ionocyte, was revealed (*Plasschaert et al., 2018*). As *CFTR* is the gene mutated in cystic fibrosis, cell-specific therapies for this disease can now be evaluated.

In our mice exposed to IH, prostacyclin synthase (*Ptgis*) expression was dramatically upregulated in myofibroblasts and fibroblasts. As a potent vasodilator and inhibitor of platelet aggregation, therapies targeting this pathway are already used for patients with pulmonary arterial hypertension (*Farber and Gin-Sing, 2016*). As another example, fibroblast growth factor receptor 2 (*FGFR2*) is highly expressed in AT2 cells. FGFR2 promotes alveolar regeneration in response to lung injury (*Perl and Gale, 2009*) and is upregulated in patients with IPF (*Li et al., 2018*). Previous studies suggest that OSA can induce injury to the lung (*Aihara et al., 2011*; *Lederer et al., 2009*), and OSA is prevalent in patients with IPF (*Lancaster et al., 2009*). If OSA does, in fact, lead to fibrotic changes in the lung, targeting FGF pathways in alveolar epithelial cells could prevent disease progression from IH.

Although there is limited data on the role of the circadian clock in OSA (*von Allmen et al., 2018*; *Yang et al., 2019*; *Gabryelska et al., 2020*), it may play a significant role in this disease (*Entzian et al., 1996*; *Smith et al., 2017*; *Butler et al., 2015*). We found dysregulation of circadian gene expression in multiple cell types. The circadian clock is a transcriptional/translational feedback loop that coordinates ~24 hr timing of physiological functions. BMAL1, the key clock TF (*Hogenesch et al., 1998*), interacts with CLOCK (*King et al., 1997*) and its partner NPAS2 (*Zhou et al., 1997*) to activate hundreds of target genes. All three are members of the basic helix-loop-helix (bHLH)-PER-ARNT-SIM (PAS) TF family. The HIFs (1–3) are members of the same TF family and are stabilized under low-oxygen conditions. Alterations in expression of clock genes are reported in peripheral blood cells from patients with OSA (*Yang et al., 2019*). These changes in expression in blood cells are not affected by CPAP treatment (*Moreira et al., 2017*), suggesting that they persist despite treatment. It is known that the circadian clock in alveolar epithelial cells impacts pulmonary physiology, and its disruption can contribute to disease in animal models (*Zhang et al., 2019*). More importantly, IH leads to intertissue circadian misalignment (including in the lung) in mice (*Manella et al., 2020*). The hypoxic episodes that define OSA are clearly diurnal, but we do not understand if clock disruption is a cause or consequence of disease. Our findings suggest that circadian clock dysfunction may be an important early-stage consequence of hypoxia-driven disease and may contribute to downstream processes.

Lung samples from IH-exposed mice did *not* show comprehensive histopathologic changes. This contrasts with findings from some prior murine models of IH. For example, IH induced epithelial cell proliferation in lungs (*Reinke et al., 2011a*). These models also produced other phenotypic changes, such as increased lung volumes (*Reinke et al., 2011b*). In a bleomycin-induced lung injury model, fibrosis in mouse lung is worsened by IH (*Gille et al., 2018*). In our model, short-term exposure to IH did not result in changes to the parenchyma or vessels. This is likely attributed to the difference in the length of time of IH exposure. We exposed our mice to IH for a shorter period of time to specifically evaluate the changes in gene expression prior to comprehensive lung remodeling with the hope of uncovering early pathways that lead to disease. Based on the shorter time of exposure, we are likely seeing molecular changes from IH with minimal gross architectural changes to the lung. Future longitudinal studies should address how gene expression profiles change over time and which cell types drive disease progression from early to late IH exposure. Early insults from hypoxia may also drive organ-specific damage in other systems. Although models of IH replicate desaturation and recovery of fractional inhaled oxygen, there is variability in the number of hypoxic events, length of desaturation events, and length of overall exposure. This variability could affect expression profiles and histopathologic findings.

Several different animal models have been designed for the study of OSA and those associated pathophysiologic sequelae, each with its own advantages and disadvantages. For example, early studies using animal models made use of implanted balloons (*Crossland et al., 2013*). The most widely used model, implemented here, uses cyclic episodes of IH with recovery to room air during inactive phases (*Trzepizur et al., 2018*). Both models can produce vascular and metabolic disturbances associated with OSA, such as sympathetic dysregulation (*Chalacheva et al., 2013*; *Ferreira et al., 2020*), changes in blood pressure (*Crossland et al., 2013*; *Schulz et al., 2014*), and metabolic dysregulation (*Li et al., 2005*; *Jun et al., 2014*). Chronic exposure to IH has also been

shown to induce lung growth in adult mice (*Reinke et al., 2011b*). Models using IH allow for the study of oxygen desaturations without the need for surgical implantation or scarring in the airway.

Delineating the upstream processes dysregulated in OSA could help us identify potential candidates for therapeutic intervention. Given the socioeconomic burden to our healthcare system for diagnosing and treating OSA, new diagnostic and therapeutic strategies will be vital for the coming years.

# Materials and methods

## Key resources table

| Reagent type (species) or resource | Designation | Source or reference | Identifiers | Additional information |
|---|---|---|---|---|
| Strain, strain background (*Mus musculus*, C57BL/6J) | Wild-type C57BL/6 (WT) | Jackson Laboratories | RRID: IMSR_JAX:026133 | See Materials and methods |
| Antibody | Anti-FOXF1 (Goat Polyclonal) | R and D Systems | Cat# AF4798, RRID:AB_2105588 | IF (1:100) |
| Antibody | Anti-Lyve1 (Rabbit Polyclonal) | Abcam | Cat# AB14917, RRID:AB_301509 | IF (1:100) |
| Antibody | Anti-actin, smooth muscle (Mouse Monoclonal IgG2A) | Sigma–Aldrich | Cat# A5228, RRID:AB_262054 | IF (1:2000) |
| Antibody | Anti-Ki-67 (Monoclonal Mouse IgG1) | BD Biosciences | Cat# 556003, RRID:AB_396287 | IF (1:100) |
| Antibody | Anti-SFTPC (Rabbit Polyclonal) | Seven Hills Bioreagents | Cat# WRAB-9337, RRID:AB_2335890 | IF (1:1000) |
| Antibody | Anti-Periostin (Rabbit Polyclonal) | Abcam | Cat# ab215199 | IF (1:100) |
| Antibody | Anti-Endomucin (Goat Polyclonal) | R and D Systems | Cat# AF4666, RRID:AB_2100035 | IF (1:200) |
| Antibody | Anti-SOX9 (Rabbit Polyclonal) | Millipore Sigma | Cat# AB5535, RRID:AB_2239761 | IF (1:100) |
| Antibody | Anti-Hop (E1), (Monoclonal Mouse IgG1) | Santa Cruz Biotechnology | Cat# SC_398703, RRID:AB_2687966 | IF (1:100) |
| Commercial assay or kit | Collagenase, elastase, dispase digestion buffer | Sigma-Aldrich; Worthington Biochemical | | |
| Commercial assay or kit | Bacillus licheniformis mix | Sigma-Aldrich | | |
| Software, algorithm | Imaris (Bitplane) | https://imaris.oxinst.com/ | RRID:SCR_007370 | Version 9.6 |
| Software, algorithm | Nikon NIS-Elements software | https://www.microscope.healthcare.nikon.com/en_EU/products/software/nis-elements | RRID:SCR_014329 | |
| Software, algorithm | STAR | *Dobin et al., 2013* | RRID:SCR_015899 | Version 2.5 |
| Software, algorithm | HTSeq | *Anders et al., 2015* | RRID:SCR_005514 | Version 0.6.0 |
| Software, algorithm | Cell Ranger | https://support.10xgenomics.com/single-cell-gene-expression/software/downloads/latest | RRID:SCR_017344 | Version 2.1.1 |
| Software, algorithm | AltAnalyze | http://www.altanalyze.org/ | RRID:SCR_002951 | Version 2.1.2 |
| Software, algorithm | DESeq2 | *Love et al., 2014* | RRID:SCR_015687 | Version 1.24.0 |

*Continued on next page*

*Continued*

| Reagent type (species) or resource | Designation | Source or reference | Identifiers | Additional information |
|---|---|---|---|---|
| Software, algorithm | cellHarmony | *DePasquale et al., 2019* | | |
| Software, algorithm | drugBankR | https://github.com/yduan004/drugbankR; *Duan, 2019* | | |
| Software, algorithm | GOSemSim | *Yu et al., 2010* | | Version 2.10.0 |
| Software, algorithm | SINCERA | *Guo et al., 2015* | RRID:SCR_016563 | Version 0.99.0 |
| Software, algorithm | CellPhoneDB | *Vento-Tormo et al., 2018* | RRID:SCR_017054 | |
| Other | DisGeNET | http://www.disgenet.org | RRID:SCR_006178 | Version 6.0 |
| Other | DrugBank | http://www.drugbank.ca/ | RRID:SCR_002700 | Version 5.1.4 |

## Animals

Use of animals and all procedures were approved by the Institutional Animal Care and Use Committee at Cincinnati Children's Hospital Medical Center and complied with the National Institutes of Health guidelines. Male C57BL/6J wild-type mice (RRID: IMSR_JAX:026133) aged 6 weeks were purchased from The Jackson Laboratory (Bar Harbor, ME) and entrained to a 12 hr:12 hr light:dark cycle for 2 weeks prior to exposure.

## Experimental design

All studies were conducted in 8–10 weeks old male mice. Mice were housed in light boxes and entrained to a 12:12 light:dark cycle for 2 weeks prior to initiation of IH. Mice were randomly assigned to IH or room air exposures. For the experimental group, mice were maintained in a commercially designed gas control delivery system (Model A84XOV, BioSpherix, Parish, NY) during the inactive (light) phase from (ZT 0–12). Mice were provided with food and water ad libitum. For each episode of IH, the fractional inhaled oxygen ($O_2$) was reduced from 20.9% to 6% over a 50 s exposure period, followed by an immediate 50 s recovery period to 20.9%. The fractional oxygen was maintained at 20.9% for approximately 15 s before the cycle was repeated, allowing for ~30 hypoxic events per hour. Ambient temperature in the hypoxia chamber was maintained between 22 and 24°C to match room air. Mice in the experimental group were maintained at room air during the active phase, ZT 13–24. Mice in the control group were maintained at room air throughout the circadian cycle, ZT 0–24. Experimental and control mice were exposed to IH vs room air for 9 days, followed by sacrifice at ZT 3 on day 10 of exposure. This was immediately followed by organ harvest and preparation for bulk RNA-seq, scRNA-seq, or histopathology.

Animal models of IH are performed by exposing mice to cycling levels of room air (fraction of inspired oxygen [$FiO_2$] – 21%) to different trough/nadir values (most commonly from 5 to 15%), producing dose–response effects (*Nagai et al., 2014*; *Lim et al., 2016*; *Gallego-Martin et al., 2017*; *Farré et al., 2018*). These fluctuations in $FiO_2$ translate to measured oxygen saturation ($SaO_2$) nadir values of approximately 50–70% in mice (*Jun et al., 2010*; *Lim et al., 2015*; *Reinke et al., 2011b*). These cycling $SaO_2$ nadirs in mice closely correspond to partial pressure of oxygen ($PaO_2$) values seen in humans with OSA (*Farré et al., 2018*). $PaO_2$ levels in humans are lower than for mice for any given $SaO_2$ value (*Farré et al., 2018*). Hence, $SaO_2$ nadir values that commonly occur in patients with OSA (70–90%) correspond to $SaO_2$ nadir values (40–70%) in mice exposed to our cycling $FiO_2$ trough levels. Based on the diagnostic criteria for severe OSA in humans (*Kapur et al., 2017*), our protocol for 30 hypoxic events an hour with cycling $FiO_2$ troughs of 6% in mice represent clinically severe OSA in humans.

## RNA isolation

In total, six experimental and six control mice were used for bulk RNA-seq. For bulk RNA-seq, the lung was quickly harvested and snap-frozen in liquid nitrogen. Organs were later homogenized in TRIzol reagent (Invitrogen) and processed using a bead mill homogenizer (Qiagen Tissuelyser). RNA was then isolated from lung homogenates by phase separation using chloroform and phase

separation columns. The aqueous phase was then applied to an RNeasy column following the manufacturer's protocol (Qiagen) to extract and purify the RNA.

## Bulk RNA sequencing and analysis

RNA from the lungs of control and experimental mice were sent for bulk sequencing separately. Approximately 0.4 μg of total RNA was used for library preparation. mRNA enrichment and library preparation were performed using the Polyadenylated (PolyA+) mRNA Magnetic Isolation Module (New England Biolabs) and NEBNext Ultra II RNA Library Prep Kit for Illumina (New England Biolabs), following the manufacturer's protocol. All 12 samples were then pooled together and sequenced in one lane using Illumina Novaseq 6000 platform with paired-end 150 bp (*Supplementary file 1a*). The raw fastq files from RNA-seq were mapped to GRCm38 mouse genome reference using STAR (version 2.5) with default parameters. More than 90% (*Supplementary file 1a*) of sequenced paired-end reads (above 50M reads for each library) were mapped to the mouse genome by STAR (*Dobin et al., 2013*; RRID:SCR_015899). HTSeq (*Anders et al., 2015*; RRID:SCR_005514) (version 0.6.0) was used to quantify gene expression, with Ensembl GRCm38.96 as a reference. DESeq2 (RRID:SCR_015687; version 1.24.0) was used to perform the differential expression analysis on the HTSeq quantified count per gene. The top 200 up- and downregulated genes (*Supplementary file 1b*), ranked by p-value from low to high with fold change above 1.5 (or log2[fold change] > 0.58), were used for biological process enrichment analysis in the DAVID database. Biological process terms with at least five differentially expressed genes and BHQ < 0.15 were selected. For aggregating redundant biological processes, GOSemSim (*Yu et al., 2010*) (version 2.10.0) was used to calculate the semantic similarity ('Jiang' method from GOSemSim) between significant biological processes. Redundant biological processes were manually merged into biological process categories (*Supplementary file 1c*).

## Dissociation protocol for single-cell sequencing

For scRNA-seq, a total of three biological replicates (three IH and three controls in the first experiment and two of each in the other replicates). On the last day of exposure, the mice were sacrificed, lung harvested, and tissue immediately placed in ice-cold phosphate-buffered saline (PBS). Dissociation of the pooled whole mouse lung for each group (IH vs control) was performed as previously described (*Guo et al., 2019*). Briefly, minced lung was placed in collagenase/elastase/dispase digestion buffer (Sigma–Aldrich, St. Louis, MO; Worthington Biochemical, Lakewood, NJ). After mixing on ice for approximately 3 min, the lung was minced again. After resting the suspension, the supernatant was passed through a 30 μM filter. A *Bacillus licheniformis* mix (Sigma–Aldrich) was added to the cell suspension, mixed on ice for approximately 10 min, and passed through a 30 μM filter. The suspension was spun at 500 g for 5 min at 4°C. The pellet was rinsed with a red blood cell lysis buffer. This was again passed through a 30 μM filter and then spun at 500 g for 5 min at 4°C. The cell suspension was resuspended in PBS/bovine serum albumin and manually counted with a hemocytometer. The volume was adjusted to obtain a final concentration of approximately 1000 cells/μl to be loaded to the 10× Chromium platform.

## scRNA-seq library construction and sequencing

The single-cell suspension was applied to the 10× Genomics Chromium platform (San Francisco, CA) to capture and barcode cells, as described in the manufacturer's protocol. Libraries were constructed using the Single Cell 3′ Reagent Kit (v2 Chemistry). The completed libraries were then sequenced using HiSeq 2500 (Illumina, San Diego, CA) running in Rapid Mode. Each sample was loaded onto two lanes of a Rapid v2 flow cell.

### scRNA-seq data processing

Raw data from 10× Genomics were demultiplexed and converted to a fastq file using cellRanger (RRID:SCR_017344; v2.1.1) mkfastq. Reads from the same library sequenced in different flow cells (technical replicates) were combined and aligned to the mm10 genome reference using cellRanger count. Summary data for statistical mapping profiles are presented in *Supplementary file 2a*. The gene expression profiles for cells from the three biological replicates of the IH group were combined with cellRanger aggr and were run an unsupervised analysis using the software Iterative Clustering

and Guide-gene Selection (ICGS) versions 2 (AltAnalyze version 2.1.2) to generate reference clusters using the program defaults with Euclidean clustering (*DePasquale et al., 2019*). ICGS2 grouped 12,324 cells into 25 reference clusters based on the expression profiles of 1480 selected marker genes (*Supplementary file 2b*). All cells from control and IH groups were then aligned to these 25 reference clusters using cellHarmony (*DePasquale et al., 2019*). Uniform Manifold Approximation and Projection (UMAP) calculation was run using integrated function in AltAnalyze (RRID:SCR_ 002951) -v2.1.2 with default parameters. For annotating the 25 reference clusters into known lung cell types, we prepared a comprehensive marker gene list for known lung cell types. The sources of this marker gene list included information from the Mouse Cell Atlas, ToppGene, and Lung Gene Expression Analysis (LGEA) (*Tabula Muris Consortium et al., 2018*; *Chen et al., 2009*; *Du et al., 2017*). Additionally, we manually collected cell marker genes from published scRNA-seq studies performed in mouse or human lung (*Zilionis et al., 2019*; *Guo et al., 2019*). One-tailed Fisher's exact test was used to perform enrichment analysis between marker genes for each cluster and the curated reference markers of known lung cell types. Each cluster was manually assigned to a specific cell type based on the known cell type with the lowest BH (*Benjamini and Hochberg, 1995*) adjusted p-value (GO-Elite software) (*Zambon et al., 2012*). Those clusters corresponding to the same annotated cell type were manually joined as one cell type for downstream analyses (e.g. endothelial corresponding to four clusters). This process reduced the 25 reference clusters into 19 cell types (*Supplementary file 2c*). For testing the cell-type composition difference of mouse lung between experimental and control groups, a centered log ratio transformation was performed on the percentage of each cell type before applying the t-test (two tailed). The statistical p-value from the t-test was adjusted with the BH method.

## Pseudo-bulk RNA-seq differential expression analysis

To identify differentially expressed genes in each lung cell type between the control and IH groups with multiple biologic replicates, all cells assigned to the same cell type were aggregated into a 'pseudo-bulk' data library by library. For each library, the sum of the reads per gene from cells assigned to the same cell type were used to represent the cell-type-specific gene expression profiles. Percentage of cells expressed per gene were calculated as a fraction of cells with $\geq$1 read(s) for the gene in each cell type. Count per one million UMI (CPM) for each cell type was calculated as the (sum of reads per gene/sum of reads) * 1,000,000 for each library. Differential expression analysis was performed with DESeq2 for each cell type, using the sum of reads per gene as input, with three replicates in each of the control and IH groups. Ranked by p-value from low to high, the top 200 up- and downregulated genes (*Supplementary file 2d*) with fold change above 1.2 (or log2[fold change] > 0.26) were used for biological process enrichment analysis in the DAVID database. Selecting and aggregating biological processes (*Supplementary file 2e*) were performed as described in the Materials and methods section labeled, 'Bulk RNA sequencing and analysis'. We further extracted those genes enriched in circadian rhythm and immune response and selected well-established genes in each biological process to demonstrate their expression variation under IH exposure in each cell type based on literature searches.

## Endothelial subpopulation analysis

To improve the accuracy for classifying endothelial cells, we reran the AltAnalyze-ICGS2 clustering algorithm using only the 5579 cells annotated to pulmonary vascular endothelial cells from IH-exposure groups, followed by a realignment of all vascular endothelial cells from both control and IH groups to these clusters using cellHarmony. AltAnalyze-ICGS2 produced six clusters with 314 marker genes (*Supplementary file 3a*). We matched the ICGS2-selected marker genes with vascular endothelial subpopulation marker genes presented in a study from *Travaglini et al., 2019*. This annotated the clusters into four subpopulations of endothelial artery, vein, capillary aerocytes, and general capillary cells (*Supplementary file 3b*). We further aggregated all cells of the same vascular endothelial subpopulations into 'pseudo-bulk' data library by library. DESeq2 was used to detect differentially expressed genes for each subpopulation between the control and IH-exposure groups. To select differentially expressed genes in each endothelial subpopulation, the cutoff was set as BHQ < 0.2 and fold change > 1.2 (or log2[fold change] > 0.26). Those differential expression genes (*Supplementary file 3c*) in endothelial capillary cells were used for BP enrichment analysis in the

DAVID database. Selecting and aggregating biological processes (*Supplementary file 3d*) were performed as described in the Materials and methods section labeled, 'Bulk RNA sequencing and analysis'. Transcription factor analysis for capillary aerocytes and general capillary cells were run using SINCERA (*Guo et al., 2015*; RRID:SCR_016563). The analysis for control vs IH-exposure groups were run separately. All cells from the three biological replicates were used for a combined analysis. Cell annotations from AltAnalyze were used to select cells annotated to aerocytes and general capillary cells, respectively. For each cell cluster, we selected for marker genes using the FindMarkers function in Seurat (*Satija et al., 2015*) and filtered with parameters pct.1 $\geq$ 0.2, avg_log2FC $\geq$ log(1.5), p_val_adj < 0.1. The filtered marker genes were used as the target genes for driving force analysis in SINCERA. The predicted key TF list for aerocytes and general capillary cells for control and IH-exposure groups are shown in *Supplementary file 3e, f*. A network regulatory figure is plotted using igraph (*Csardi and Nepusz, 2006*) package in R. The top 25 predicted TFs from control and IH-exposure groups were selected.

## Association analysis on IH-responsive genes with pulmonary disease genes and drug targets at the cell-type level

Gene–disease association information was downloaded from the DisGeNET (*Piñero et al., 2017*; RRID:SCR_006178) database (curated gene–disease associations). The downloaded file was filtered with keywords to specifically select genes linked to pulmonary diseases, which includes 'allergic asthma', 'asthma', 'chronic obstructive airway disease', 'pulmonary hypertension', 'chronic thrombo-embolic pulmonary hypertension', 'idiopathic pulmonary arterial hypertension', 'familial primary pulmonary hypertension', 'idiopathic pulmonary hypertension', 'interstitial lung fibrosis', 'pulmonary fibrosis', and 'idiopathic pulmonary fibrosis'. One-tailed Fisher's exact test was used to determine whether the top 200 up- or downregulated genes in each cell type exposed to IH significantly overlapped with these pulmonary disease-associated genes (*Supplementary file 4a*). The significant cutoff was set with a BH adjusted p-value from a Fisher's exact test of <0.05 and at least five overlapped genes with any pulmonary disease gene set. For the association analysis between IH-responsive genes and pulmonary drug targets, the xml file was downloaded from the DrugBank (*Wishart et al., 2008*; RRID:SCR_002700) (version 5.1.4). The drugbankR package (https://github.com/yduan004/drugbankR; *Duan, 2019*) was used to parse the xml file to get each drug and its target genes. The parsed drug table was linked with the top 200 up- or downregulated genes in each cell type exposed to IH by drug target genes. The drug table was further filtered with respiratory tissue and disease-associated key words (e.g. asthma, lung, bronchus, airway, etc.) to keep candidate drugs used to treat pulmonary diseases. The filtered table was manually curated to select drugs mainly indicated to treat pulmonary diseases. In the association analysis, the 'homologene' package (https://github.com/oganm/homologene; *Mancarci, 2019*) was used to find the human-unique homolog of mouse genes. CellPhoneDB (RRID:SCR_017054) is used to predict ligand–receptor interactions between cell types. The pulmonary disease-relevant drug-targeted gene list (*Supplementary file 4b*) is used to select ligand–receptor interaction pairs. Genes with unknown drug effects were filtered out from the analysis. Human homolog gene is used for this analysis. Gain of interactions and loss of interactions are calculated by summing up the number of ligand–receptor interactions that were only significantly present in either hypoxia or control (p-value<0.05 and mean>0.1), respectively, for each cell pair comparison. Two-tailed Fisher's exact test was used to calculate significance changes of each cell type. Hive plot is made using the HiveR package (https://github.com/bryanhanson/HiveR; *Hanson, 2017*). The gene set enrichment analysis (GSEA; *Subramanian et al., 2005*) was performed for myofibroblasts (*Supplementary file 4c*). The expressed genes in myofibroblasts were ranked by the fold change calculated from pseudo-bulk expression values in control and IH-exposure samples. The 'homologene' package was used to find the human-unique homolog of correlated mouse genes. The canonical pathway list was downloaded from MSigDB (http://www.gsea-msigdb.org/gsea/msigdb/collections.jsp). The fgsea R package (*Korotkevich et al., 2016*) was used to run GSEA on the ranked genes in myofibroblasts. The gene sets were restricted to REACTOME subsets with at least 10 genes but less than 500 genes. The 'nperm' was set to 100,000. The significantly enriched gene sets were selected by padj<0.15 and absolute NES values above 1.7. With this cutoff value, no gene sets with downregulated leading edge genes after exposure to IH are significantly enriched. The predicted key TFs for myofibroblasts (*Supplementary file 4d*) were performed in a similar way as with endothelial subpopulations. To

highlight the IH-driven key TFs, the top 25 TFs predicted from lung myofibroblasts after IH exposure were compared to all significant TFs predicted from lung myofibroblasts in control samples.

## Lung fixation, histological staining, immunofluorescence, confocal microscopy, and quantification of confocal images

For histological staining, mice were sacrificed, and lung inflation fixation was immediately performed. After exposure of the trachea and lungs, the trachea was cannulated with a scalp vein cannula (EXELINT, Redondo Beach, CA), and 10% neutral buffered formalin was gravity perfused into the lung at a height of 25 cm. After infusion, the lung was harvested and placed in formalin for 24 hr. Whole lung was then dehydrated in 70% ethanol and embedded in paraffin. For morphologic evaluation, 5 μm thick sections were cut from the paraffin blocks and stained with hematoxylin and eosin.

Immunofluorescence staining on 10% formalin-fixed mouse lung was performed on 5 μm thick paraffin-embedded tissue sections. Tissue slides were melted at 60°C for 2 hr, following rehydration through xylene and alcohol, and finally in PBS. Antigen retrieval was performed in 0.1 M citrate buffer (pH 6.0) by microwaving. Slides were blocked for 2 hr at room temperature using 4% normal donkey serum in PBS containing 0.2% Triton X-100 and then incubated with primary antibodies diluted in blocking buffer for approximately 16 hr at 4°C. Primary antibodies included ACTA2 (1:2000, Sigma–Aldrich; RRID:AB_262054), EMCN (1:200, R and D Systems; RRID:AB_2100035), FOXFI (1:100, R and D Systems; RRID:AB_2105588), HOPX (1:100, Santa Cruz Biotechnology; RRID: AB_2687966), MKI67 (1:100, BD Biosciences; RRID:AB_396287), POSTN (1:100, ABCAM), Pro-SFTPC (1:1000, Seven Hills Bioreagents; RRID:AB_2335890), SOX9 (1:100, Millipore; RRID:AB_2239761), and LYVE1 (1:100, ABCAM; RRID:AB_301509). Secondary antibodies conjugated to Alexa Fluor 488, Alexa Fluor 568, or Alexa Fluor 633 were used at a dilution of 1:200 in blocking buffer for 1 hr at room temperature. Nuclei were counterstained with DAPI (1 μg/ml) (ThermoFisher). Sections were mounted using ProLong Gold (ThermoFisher) mounting medium and coverslipped. Tissue sections were then imaged on an inverted Nikon A1R confocal microscope using a NA 1.27 objective using a 1.2 AU pinhole. Maximum-intensity projections of multi-labeled Z-stack images were generated using Nikon NIS-Elements software (RRID:SCR_014329).

Heterogeneous cell populations from paraffin-embedded immunofluorescent confocal images (60× magnification) were characterized in Imaris (Bitplane; RRID:SCR_007370), version 9.6. An average of 185 cells from control mice and 301 cells from IH-treated mice (n = 4 mice, six images per mouse) were counted specifically for proliferating cells (MKI67), endothelial cells (FOXF1), and alveolar type II cells (SFTPC) using spot detection or manual counting of cells. The cell percentage endothelial cells was calculated as the ratio between cells stained with FOXF1 and totally counted nucleus stained with DAPI in each image. Similarly, we got the cell percentages of AT2 and proliferating cells. All the remaining cells without FOXF1, SFTPC, and MKI67 staining or a cell stained with both MIK67 and SFTPC were taken as one composition group in the composition difference analysis. For testing the composition difference of endothelial, AT2, and proliferating cells in mouse lung between experimental and control groups, a centered log ratio transformation was performed on the percentage of each cell type before applying the t-test (two tailed). The statistical p-value from the t-test was adjusted with the BH method.

Alveolar wall thickness was quantified in Imaris (Bitplane), version 9.6, using the measurement tool for autofluorescence to define the alveolar walls. A total of 20 alveolar wall thickness measurements were taken for each 60× magnification confocal image (n = 2 mice per treatment, six images per mouse). Alveolar area and MaxFeret90 area were quantified in Nikon Elements, version 4.5. Maxferet90 is the distance measured orthogonally to the maximum feret chord. The larger Maxferet90 value indicates more openness in the alveolus. General analysis was run on maximum-intensity projection confocal images (20× magnification), n = 2 mice for control- and IH-treated mice, with three images per mouse. Autofluorescence with the TRITC channel was used to define alveolar areas. Thresholding was turned on, and binary processing was created for inverting, removing objects touching borders, and filtering an object area. Airways, arteries, veins, and alveolar ducts were filtered out using a minimum value of 50 and a maximum value of 4000 using the filter on object area binary. Feature area and MaxFeret90 values were exported using Microsoft Excel.

## Acknowledgements

We thank Kathryn A Wikenheiser-Brokamp for her thoughtful discussion and guidance. We thank Bruce J Aranow, Minzhe Guo, Emily R Miraldi, Yan Xu, Kashish Chetal, and J Matthew Kofron for their thoughtful discussion about this manuscript. We would like to thank S Steven Potter for allowing us to use resources available in his lab. We would also like to thank Kalpana Srivastava for work on sectioning tissue specimens and completing immunofluorescence assays.

## Additional information

### Funding

| Funder | Grant reference number | Author |
|---|---|---|
| National Institutes of Health | 5K08HL148551-02 | David F Smith |
| American Laryngological, Rhinological and Otological Society | 2017 Career Development Award | David F Smith |
| American Society of Pediatric Otolaryngology | 2016 Basic Research Award | David F Smith |
| Cincinnati Children's Hospital Medical Center | Procter Scholar Award | David F Smith |

The funders had no role in study design, data collection and interpretation, or the decision to submit the work for publication.

### Author contributions

Gang Wu, Yin Yeng Lee, Conceptualization, Data curation, Software, Formal analysis, Validation, Methodology, Writing - original draft, Writing - review and editing; Evelyn M Gulla, Conceptualization, Data curation, Validation, Methodology; Andrew Potter, Data curation, Formal analysis, Methodology, Writing - review and editing; Joseph Kitzmiller, Conceptualization, Software, Validation, Investigation, Methodology, Writing - review and editing; Marc D Ruben, Conceptualization, Formal analysis, Methodology, Writing - original draft, Writing - review and editing; Nathan Salomonis, Resources, Software, Formal analysis, Methodology, Writing - review and editing; Jeffery A Whitsett, Conceptualization, Resources, Software, Methodology, Writing - review and editing; Lauren J Francey, Conceptualization, Resources, Data curation, Formal analysis, Validation, Investigation, Methodology, Writing - original draft, Project administration, Writing - review and editing; John B Hogenesch, Conceptualization, Resources, Software, Supervision, Funding acquisition, Investigation, Methodology, Writing - original draft, Project administration, Writing - review and editing; David F Smith, Conceptualization, Resources, Data curation, Supervision, Funding acquisition, Validation, Investigation, Methodology, Writing - original draft, Project administration, Writing - review and editing

### Author ORCIDs

Gang Wu  https://orcid.org/0000-0002-4526-2034
Yin Yeng Lee  https://orcid.org/0000-0002-8092-3926
David F Smith  https://orcid.org/0000-0002-0048-4012

### Ethics

Animal experimentation: This study was performed in strict accordance with the recommendations in the Guide for the Care and Use of Laboratory Animals of the National Institutes of Health. All of the animals were handled according to approved institutional animal care and use committee (IACUC) protocols (#2019-0028) of the Cincinnati Children's Hospital Medical Center.

### Decision letter and Author response

Decision letter https://doi.org/10.7554/eLife.63003.sa1
Author response https://doi.org/10.7554/eLife.63003.sa2

## Additional files

### Supplementary files

• Supplementary file 1. This file contains tables a–c (referenced in Materials and methods). (a) The alignment statistics of bulk RNA-seq data for IH exposure and control mice. (b) Top 200 up- and downregulated genes in mice lung under IH exposure from the bulk RNA-seq data. (c) DAVID-enriched biological processes and merged categories using top 200 differential expression genes from the bulk RNA-seq data.

• Supplementary file 2. This file contains tables a–e (referenced in Materials and methods). (a) The alignment statistics of scRNA-seq data for IH exposure and control mice. (b) The marker genes list for 25 clusters from AltAnalyze analysis of scRNA-seq data from IH exposure mice. (c) The annotated cell types for 25 cellHarmony aligned clusters of scRNA-seq data from IH exposure and control mice. (d) Top 200 up- and downregulated genes in 19 detected lung cell types under IH exposure from the scRNA-seq data. (e) DAVID-enriched biological processes and merged categories using top 200 differential expression genes in each lung cell type under IH exposure.

• Supplementary file 3. This file contains tables a–f (referenced in Materials and methods). (a) The marker genes list for six clusters from AltAnalyze analysis of scRNA-seq data from annotated vascular endothelial cells of IH exposure mice. (b) The annotated endothelial subpopulations for six cellHarmony aligned clusters of scRNA-seq data from annotated vascular endothelial cells of IH exposure and control mice. (c) Differential expression genes of four annotated endothelial subpopulations under IH exposure from the scRNA-seq data. (d) DAVID-enriched biological processes and merged categories using differential expression genes of lung capillary cells under IH exposure. (e) Ranking of predicted transcription factors in capillary aerocytes using SINCERA. (f) Ranking of predicted transcription factors in general capillary cells using SINCERA.

• Supplementary file 4. This file contains tables a–d (referenced in Materials and methods). (a) The pulmonary disease-associated genes from top 200 up- or downregulated genes in each detected lung cell type exposed to IH. (b) The drug table linking drugs mainly indicated to treat pulmonary diseases and top 200 up- or downregulated genes in each cell type exposed to IH. (c) Gene set enrichment analysis of expressed genes in myofibroblasts in mice lung exposed to IH compared to controls. (d) Ranking of predicted transcription factors in myofibroblasts using SINCERA.

• Transparent reporting form

### Data availability

Sequencing data has been uploaded to GEO (GSE145436), as mentioned in the manuscript 'Data and Materials Availability' section.

The following dataset was generated:

| Author(s) | Year | Dataset title | Dataset URL | Database and Identifier |
|---|---|---|---|---|
| Wu G, Yin Yeng L, Gulla EM, Potter A, Kitzmiller J, Ruben MD, Salomonis N, Whitsett JA, Francey LJ, Hogenesch JB, Smith DF | 2020 | Short-term exposure to intermittent hypoxia in mice leads to changes in gene expression seen in chronic pulmonary disease | https://www.ncbi.nlm.nih.gov/geo/query/acc.cgi?acc=GSE145436 | NCBI Gene Expression Omnibus, GSE145436 |

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
