## [Decision Letter]

**Acceptance summary:**

Obstructive sleep apnea is an important medical problem, with elevated cardiovascular risk as a common association. Intermittent hypoxic episodes are a good predictor of such risk. The authors use single cell genomics to delineate the changes in various lung cell types in intermittent hypoxia models, with potential mechanistic insights that are translatable to OSA and other lung diseases with hypoxia as a core component.

**Decision letter after peer review:**

Thank you for submitting your article "Short-term exposure to intermittent hypoxia leads to changes in gene expression seen in chronic pulmonary disease" for consideration by *eLife*. Your article has been reviewed by three peer reviewers, including Anurag Agrawal as the Reviewing Editor and Reviewer #1, and the evaluation has been overseen by Matt Kaeberlein as the Senior Editor. The following individual involved in review of your submission has agreed to reveal their identity: U Mabalirajan (Reviewer #2).

The reviewers have discussed the reviews with one another and the Reviewing Editor has drafted this decision to help you prepare a revised submission.

Summary:

Obstructive sleep apnea is an important medical problem, with elevated cardiovascular risk as a common association. Intermittent hypoxic episodes are a good predictor of such risk so a connection is indeed plausible. The authors use single cell genomics to delineate the changes in intermittent hypoxia models, with interesting insights, but what limits enthusiasm is validation of some hypothesis generating findings from single cell data, limiting potential mechanistic insights that are translatable to OSA

Essential revisions:

1) Since intermittent hypoxia is not identical to OSA but is a potentially useful model, the first group of essential revisions pertains to new analysis of your existing data.

a) Relevance of gene expression findings should be verified in available public data and compared to publications.

b) Compensatory mechanisms for hypoxia, including angiogenesis, are expected but produce little insight unless placed in context of disease processes relevant to OSA.

c) Whether the gene expression change can be explained by changes in cell numbers should be clearly explained. Given the complexity of information obtained, I think it warrants a more detailed analysis. It would be helpful if the authors could distil the very large volumes of information into a more extensive discussion of their findings (particularly discussing the figures in more detail).

d) Morphometry should be used in histological studies

2) Additional data is needed to increase confidence in reported findings:

a) Cell types with gross alterations should be specifically checked, including by quantitative sorting and population analysis with additional markers, to ensure cell type specificity.

b) The changes in myofibroblasts is unexpected and interesting. This needs specific verification and deeper explanation. While it is known that single cell sequencing had indicated the possible presence of new cell types, it should not ignore the already well known cell types.

c) It is really surprising to see the predominant presence of endothelial cells. This is different from available literature based on single cell sequencing based molecular cell atlas. In general, *Sox17*, a marker of endoderm, is also expressed by other endoderm derived derivatives like epithelia. (Park et al., Am J Respir Cell Mol Biol. 2006 Feb;34(2):151-7). Amine oxidase C3 is relatively new marker of myofibroblasts (Hsia et al., Proc Natl Acad Sci U S A. 2016 Apr 12;113(15):E2162-71). But this ectoenzyme also expressed abundantly in adipocytes, endothelial cell and other cells.

---

## [Author Response]

Essential revisions:1) Since intermittent hypoxia is not identical to OSA but is a potentially useful model, the first group of essential revisions pertains to new analysis of your existing data.a) Relevance of gene expression findings should be verified in available public data and compared to publications.

We agree with the reviewers. In addition to our original analyses, we have comprehensively compared gene expression findings to additional publications described in the revised manuscript. Please see the section entitled, “Pulmonary disease-regulated genes provide clinical implications for OSA at the cell-specific level” in the Results for details (see citations and links for websites).

b) Compensatory mechanisms for hypoxia, including angiogenesis, are expected but produce little insight unless placed in context of disease processes relevant to OSA.

We agree with the reviewers and have revised our manuscript throughout. For example, we added context for angiogenesis in the setting of OSA in the Results section titled, “Short-term exposure to intermittent hypoxia reshapes circadian and immune pathways in the lung.” We also expanded our description of down-regulated pathways and correlated these to OSA. For example, we expanded the section on immune response in OSA and associated changes after CPAP treatment in the Results section titled, “Diverse expression pathways were up and downregulated in the presence of intermittent hypoxia.”

c) Whether the gene expression change can be explained by changes in cell numbers should be clearly explained.

The reviewers raise an important point. Here, we verified that changes in cell numbers do not influence gene expression by 3 approaches:

1) We present bulk RNA-seq data in the manuscript. Our trend in circadian and immune pathways, for example, is the same for bulk RNA-seq (compared to the scRNA-seq data). This confirms that these trends are the same using different techniques and completely different procedures.

2)In our original analysis, we normalized the data to minimize the effects of changes in cell number on gene expression. We added information in the Results to clarify this point; listed under the section, “Diverse expression pathways were up and down regulated in the presence of intermittent hypoxia.”

3) In addition, we randomly down-sampled an equal number of cells for each cell type from the control and IH groups. The down-sampling process was performed 10 times per cell type. The fold change values of the top 200 up and down regulated genes was consistent with the fold change when using all cells from each population. As shown in Author response image 1, there is a clear linear relationship of gene expression changes between using all cells and an equal number of endothelial (Endo), myofibroblast (MyoFB), AT2, macrophage (Mphage), natural killer (NK), and B cells.

Given the complexity of information obtained, I think it warrants a more detailed analysis. It would be helpful if the authors could distil the very large volumes of information into a more extensive discussion of their findings (particularly discussing the figures in more detail).

We expanded our Results section to provide more information, with “Diverse expression pathways were up and downregulated in the presence of intermittent hypoxia,” “Pulmonary vascular endothelial subpopulations show distinctive responses to intermittent hypoxia,” and “Pulmonary disease-regulated genes provide clinical implications for OSA at the cell-specific level.”

d) Morphometry should be used in histological studies

We added cell counts to the text and figures (Figure 3—figure supplement 1) and a more detailed analysis of morphometry (e.g. alveolar area and alveolar wall thickness; Figure 3—figure supplement 2). We did not see comprehensive changes in alveolar area and alveolar wall thickness based on morphometry quantification for mice exposed to IH compared to controls (Figure 3—figure supplement 2). Please see the revised Results section titled, “Short-term exposure to intermittent hypoxia did not lead to comprehensive histologic changes in the lung.” Additionally, we expanded on this in the Discussion.

2) Additional data is needed to increase confidence in reported findings:a) Cell types with gross alterations should be specifically checked, including by quantitative sorting and population analysis with additional markers, to ensure cell type specificity.

We agree and have expanded our description of the techniques to confirm cell types.

1) For clarification, we used 40-60 AltAnalyze selected markers in each cluster and annotated each cluster to a known cell type based on a comprehensive lung cell marker reference gene list collected from publicly available databases (e.g. Mouse Cell Atlas, ToppGene and LGEA) and published lung scRNA-seq data. See the following references for examples of the publicly available data:

a) Consortium et al., 2018.

b) Chen et al., 2009.

c) Du et al., 2017.

d) Cohen M et al., 2018, Cell 175(4):1031-1044.e18.

e) Zilionis et al., 2019.

2) We also used two canonical markers for each cell type for confirming the cell type specificity (revised Figure 2C). We further clarified the steps of ensuring cell-type specificity in the revised manuscript. Overall, the technique for determining cell type from scRNA-seq data from the lung is well-established (Salomonis N et al., 2019, Methods Mol Biol 1975:251-275; Guo et al., 2019).

3) We have quantified differences in cell numbers (with statistical analyses) for specific cell populations in the lung from mice exposed to IH versus controls. Specifically, we stained for and quantified those specific cell types with “gross alterations” from the scRNA-seq data (Figure 3—figure supplement 1). These data have been added to the Results, under the section titled, “Short-term exposure to intermittent hypoxia did not lead to comprehensive histologic changes in the lung.” Based on our expanded analytic techniques, as the reviewers recommended, and the additional cell counts, we are confident that our findings concerning the alterations in the proportion of cell types are accurate.

b) The changes in myofibroblasts is unexpected and interesting. This needs specific verification and deeper explanation. While it is known that single cell sequencing had indicated the possible presence of new cell types, it should not ignore the already well known cell types.

We agree. We performed a GSEA enrichment analysis for all of the detected genes from myofibroblasts and have revised and added several supplementary figures (Figure 6—figure supplements 4 and 5).We expanded our findings in the Results section titled, “Pulmonary disease-regulated genes provide clinical implications for OSA at the cell-specific level,” as well as in the Discussion.

c) It is really surprising to see the predominant presence of endothelial cells. This is different from available literature based on single cell sequencing based molecular cell atlas. In general, Sox17, a marker of endoderm, is also expressed by other endoderm derived derivatives like epithelia. (Park et al., Am J Respir Cell Mol Biol. 2006 Feb;34(2):151-7). Amine oxidase C3 is relatively new marker of myofibroblasts (Hsia et al., Proc Natl Acad Sci U S A. 2016 Apr 12;113(15):E2162-71). But this ectoenzyme also expressed abundantly in adipocytes, endothelial cell and other cells.

On average, 28.8% of our detected cells using scRNA-seq are endothelial cells in control mice (Figure 2—figure supplement 1). This is very similar to the percentage of endothelial cells (26.7%) reported in the recently published human lung atlas paper (Kyle J Travaglini et al. Nature 2020 Nov;587(7835):619).

In our revised figure (Figure 2C), we added Tek, along with *Sox17*, as an additional canonical endothelial marker gene. Additionally, we used *Aspn* and *Acta2* instead of *Aoc3* as canonical myofibroblast marker genes. Despite these modifications, the percentage of endothelial cells was unchanged, confirming the specificity of our cell population. We further clarified our cell identification technique (as described in reviewer comment 2a above). This has been added to the Results section. As the reviewer points out, there is a dual lineage-specific expression of *Sox17* during mouse embryogenesis (Choi E et al. Stem Cells 2012 30(10):2297). In the lung, *Sox17* is expressed in mesenchymal progenitors of the embryonic pulmonary vasculature and is restricted to vascular endothelial cells in the mature lung (Lange AW et al. Developmental Biology 387 (2014) 109-120).